



# Airborne Mid-Infrared Cavity enhanced Absorption spectrometer (AMICA)

Corinna Kloss[1,*], Vicheith Tan[1], J. Brian Leen[2], Garrett L. Madsen[2], Aaron Gardner[2], Xu Du[2], Thomas Kulessa[3], Johannes Schillings[3], Herbert Schneider[3], Stefanie Schrade[1], Chenxi Qiu[1], Marc von Hobe[1]

[1]Institute for Energy and Climate Research (IEK-7), Forschungszentrum Jülich GmbH, 52425 Jülich, Germany
[2]ABB Los Gatos, San Jose, U.S.A.
[3]Central Institute for Engineering, Electronics and Analytics (ZEA), Engineering and Technology (ZEA-1), Forschungszentrum Jülich GmbH, 52425 Jülich, Germany
[*]now at: Laboratoire de Physique et Chimie de l'Environnement et de l'Espace (LPC2E), Université d'Orléans, CNRS, Orléans, France

Correspondence to: Marc von Hobe (m.von.hobe@fz-juelich.de)

**Abstract.** We describe the Airborne Mid-Infrared Cavity enhanced Absorption spectrometer (AMICA) designed to measure trace gases *in situ* on research aircraft using Off-Axis Integrated Cavity Output Spectroscopy (OA-ICOS). AMICA contains two largely independent and exchangeable OA-ICOS arrangements, allowing for the simultaneous measurement of multiple substances in different infrared wavelength windows tailored to scientific questions related to a particular flight mission. Three OA-ICOS setups have been implemented to measure $OCS$, $CO_2$, $CO$ and $H_2O$ at 2050 cm$^{-1}$, $O_3$, $NH_3$ and $CO_2$ at 1035 cm$^{-1}$, and $HCN$, $C_2H_2$ and $N_2O$ at 3331 cm$^{-1}$. The 2050 cm$^{-1}$ setup has been fully characterized in the lab and successfully used for atmospheric measurements during two campaigns with the research aircraft M55-Geophysica and one with the German HALO aircraft. Nominal measurement precision is 30 ppt for $OCS$, 1 ppm for $CO_2$, 3 ppb for $CO$ and 100 ppm for $H_2O$. The 1035 and 3331 cm$^{-1}$ arrangements have only partially been characterized and are still in development. The ~100 kg instrument with a typical in-flight power consumption of about 500 VA is dimensioned to fit into one 19 inch rack typically used for deployment inside the aircraft cabin. Its rugged design and a pressurized and temperature stabilized compartment containing the sensitive optical and electronic hardware also allow for deployment in payload bays outside the pressurized cabin even at high altitudes of 20 km. A sample flow system with two parallel proportional solenoid valves of different size orifices allows for precise regulation of cavity pressure over the wide range of inlet port pressures encountered between the ground and maximum flight altitudes. Sample flow on the order of 1 SLM maintained by an exhaust-side pump limits the useful time resolution to about 2.5 s (corresponding to the average cavity flush time).

# 1 Introduction

Airborne in-situ trace gas observations are typically made at high spatial and temporal resolution and thus allow for the investigation of small and intermediate scale processes (e.g. Schumann et al., 2013). Most important trace gases possess reasonably strong absorption bands in the infrared region, so that infrared absorption spectroscopy offers a simple and straight-





forward measurement technique for many gases. Measurement sensitivity critically depends on path length, and many trace gases at atmospheric abundances can only be detected with path lengths of hundreds of meters or even several kilometres. With in-situ instruments, the long path lengths needed are often beyond those offered by common multi-pass cells (e.g. Rob-

ert, 2007;Herriott and Schulte, 1965;White, 1942) and are accessible only by using cavity enhanced methods where path lengths of many kilometres can be achieved with mirrors of sufficient reflectivity. Those methods all go back to the cavity ring down spectroscopy first described by O'Keefe and Deacon (1988). A wide variety of modifications and off-springs of this original technique has been developed and used for many different applications over the past three decades. Reviews with a historical overview of cavity enhanced spectroscopy and a comprehensive listing of available methods have been giv-

en by Paldus and Kachanov (2005) and more recently by Gagliardi and Loock (2014).

Cavity enhanced spectrometers in the near infrared and mid infrared region are commercially available for numerous trace gases. A cavity enhanced technique that is sensitive, robust and easy to implement is the Off-Axis Integrated Cavity Output Spectroscopy (OA-ICOS, Baer et al., 2002;O'Keefe, 1998;O'Keefe et al., 1999;Paul et al., 2001). OA-ICOS has become a well-established technique for ground based measurements of a wide range of trace gases, e.g. CO, $N_2O$, $CH_4$, $CO_2$ and wa-

ter isotopes (e.g. Arévalo-Martinez et al., 2013;Hendriks et al., 2008;Kurita et al., 2012;Steen-Larsen et al., 2013). OA-ICOS measurements on research aircraft have been made (Leen et al., 2013;O'Shea et al., 2013;Provencal et al., 2005;Sayres et al., 2009) but these instruments often rely on the instrument being placed inside a pressurized cabin and/or need inverters to reduce the frequency of the electrical power generated by the aircraft's engines from the typical 400 Hz down to the more common 50/60 Hz.

The Airborne Mid-Infrared Cavity enhanced Absorption spectrometer (AMICA) is a novel two-cavity airborne OA-ICOS analyser simultaneously measuring multiple trace gases. A first operational version for laboratory tests was completed in February 2016, and the first airborne deployment took place in August 2016. Since then, AMICA has evolved as corrections and upgrades were implemented based on the results from tests and deployments. Here, we describe the latest and current version of AMICA that has been optimized both in terms of reliability and performance. Where earlier laboratory and field

data from AMICA are presented, appropriate reference to differences in the setup and hardware used will be made. In Section 2 we describe the special design features that ensure AMICA can function optimally on a moving and vibrating aircraft at pressures and temperatures down to 50 hPa and -80 °C respectively. Section 2 also includes a detailed description of the implementation of OA-ICOS in AMICA. Details on data handling and analysis are given in Section 3. Realized applications to measure specific target species at certain wavelength windows in the infrared are described in Section 4 that also includes

results from laboratory tests and calibrations. Finally, in Section 5, the first airborne measurements that demonstrate AMICA's functionality, performance and potential are presented.



## 2 Instrument design and description

### 2.1 General setup and bulk characteristics

Figure 1 shows a 3D technical drawing of AMICA. It consists of two main compartments: the main "*ICOS enclosure*" con-
taining two OA-ICOS cavities and their respective laser and data acquisition hardware and electronics (Section 2.2), and an
attached "*power box*" containing electrical components converting the AC supply input voltage to various filtered DC volt-
ages (Section 2.5). A single stream of sampling air is drawn serially through two cavities, maintaining a constant pressure in
each cavity (Section 2.4). Safe aircraft deployment is ensured by a robust design and numerous features to withstand signifi-
cant vibrational stress as well as severe pressure and temperature conditions (Section 2.3), and by an electronic design and
grounding concept minimizing interference with the aircraft and other instruments (Section 2.5).

Not including aircraft specific rack or mounting hardware (Section 2.3), the dimensions are 1050 x 435 x 355 mm and the
weight is approximately 115 kg. The power consumption is up to 800 VA at start-up and during the initial warm up phase
(taking between 2 and 45 minutes depending on ambient conditions), and typically 500 VA during normal operation of the
warmed up instrument. An additional 230 VA can be passed through AMICA to supply power to a heated inlet when this is
needed (cf. Section 2.5). More detailed information with itemized weights and power characteristics is given in Table 1.

### 2.2 ICOS implementation

AMICA contains two largely independent ICOS systems in a pressurized and temperature stabilized enclosure (cf. Section
2.3). The arrangement of the different components can be seen in Figure 2. Each ICOS entity consists of a laser source (C7
in Figure 2 and Table 2), a 90° deflection mirror, a 508 mm long cavity of 48 mm inner diameter with two concave high re-
flectivity mirrors, a collimating lens and a detector (C8). The laser beam is aligned to enter the cavity slightly off-axis to
minimize sensitivity to vibrations and to avoid interference patterns resulting from cavity resonance (Paul et al., 2001). In
addition, the position of each mirror is modulated by three piezoelectric transducers (PZTs, modulated by C12). The colli-
mating lens on the cavity end opposite where the laser beam enters focuses light exiting the cavity onto the sensitive area of
the photodetector (C8).

Laser, mirrors and detectors are exchangeable to tailor the target species to relevant science questions of a particular mission.
Specific details on wavelength regions, mirrors as well as laser and detector models for the combinations implemented up to
now are given in Table 3, and more detailed descriptions are presented together with representative spectra and sensitivity
analyses for various trace gases in Section 4.

Each laser is operated by a Quantum Cascade Laser (QCL) controller board (C6a/b, in the following referred to as LTC-
1141) that offers both thermoelectric control to stabilize the laser temperature and the means to modulate the laser current. In
AMICA, the laser current is repetitively ramped over the lasing range or parts thereof to scan over a desired range of typical-
ly a few wavenumbers. At the end of each ramp, the laser is turned off to monitor the ring down decay and the dark signal at



the detector (the fitting of dark signal, ring down time and the measured spectra is described in Section 3). Depending on the necessary spectral resolution and on the light attenuation due to trace gas absorption in relation to loss at the mirrors, ramp-

ing is done at different rates. Typically, a laser is ramped between 100 and 1000 times per second, the details for each setup are given in Section 4.

The detector (C8; typically a photodiode, see Table 3 for specifics of each setup) translates the intensity of light exiting the cavity into a current signal, which is amplified and converted to voltage by a preamplifier (C9). The voltage signal is passed on to a fast AD channel (sampling rate: 100 MHz, input resistance: 240 Ω) of the corresponding LTC-1141 board (C6 a/b).

A custom firmware (see LTC-1141 application note under https://www.meerstetter.ch/customer-center/downloads/category/63-application-notes?download=573) installed on the LTC-1141 on-board microprocessor aver-ages the signal for $N_{ramps}$ ramps over a predefined acquisition time $t_{avg}$ with $N_{ramps} = t_{avg} / N_{pts}$ , where $N_{pts}$ is the number of points per ramp. The averaged signal ramps are transferred to the embedded PC (C1, a PC/104 stack consisting of 4 mod-ules) via UDP data stream (see Section 3.1). The data acquisition and processing capabilities of the LTC-1141 (C6 a/b) are

engaged efficiently in AMICA and significantly reduce computational load on the embedded PC.

## 2.3 Mechanical design

*Design of the thermally insulated ICOS enclosure*

As the largest and main compartment, the OA-ICOS enclosure provides the mechanical stability needed for aircraft opera-tion. Its housing and all structural elements are made of aircraft certified aluminium (EN-AW6061-T651), and the design

was laid out in order to withstand forces up to 10 g without plastic deformation. To inhibit corrosion and at the same time retain full electrical conductivity of the housing to minimize electromagnetic interference (EMI, cf. Section 2.5), a chromate conversion coating was applied to all parts prior to assembly.

The 12.7 mm thick side panels and bottom plates are bolted together by hexagon socket head cap screws (ISO 4762 – M5x16) tightened to 4 Nm into HELICOIL® thread inserts (M5x1.5D). For additional stability and to reduce shear forces

that could weaken the adhesive bonding (see below), stainless steel pins (Ø5x14, ISO 2338) are driven into pinholes between bolts. Two different enclosure lids were designed, one for cabin operation and one for operation exposed to ambient condi-tions. The latter one needs to withstand excess pressure of about 1000 hPa inside the enclosure (see below) and consists of a 6.35 mm (1/4") thick plate with extra enforcement rims where the thickness is doubled to 12.7 mm (1/2"). It is attached by 65 hexagon socket head cap screws (ISO 4762 – M5x16) tightened to 4 Nm into HELICOIL® thread inserts (M5x1.5D) in

the enclosure top rim secured with Nord-Lock® washers (NL5ss). In the lid used for cabin operation, two large openings are cut out and covered with sheet metal attached by 8 quick release fasteners with wing handles (Camloc, D4002 series) allow-ing for easy and quick maintenance access.





QCL lasers and photosensitive detectors used in mid infrared OA-ICOS typically need to be precisely and accurately temperature stabilized, and a good temperature stabilization of the cavities is also beneficial for good long term precision. To
ensure that all individually regulated components can be optimally stabilized with small amplitudes in temperature as well as power fluctuations, the entire enclosure is thermo-regulated to approximately 35 °C by two banks of thermoelectric coolers (TECs) sandwiched between heat sinks equipped with fans on each side (operation and regulation of the TEC assemblies is described in Section 2.5). These assemblies are bolted into the bottom plate using screws (ISO 4762 – M4x16) and sealed with a flat 4 mm thick EMI shielding gasket (Holland Shielding Systems BC) each. The enclosure walls are insulated on the
inside with polyethylene foam (ETHAFOAM, 4101 FR Polyethylene Foam, Midland, Michigan). Additional fans enhance air circulation inside the enclosure to improve temperature uniformity.

For the thermal stabilization to work efficiently and to enable the safe operation when the instrument is placed outside the aircraft cabin and thus exposed to ambient conditions at up to 20 km altitude, the enclosure is further designed to be pressure tight. To achieve this, an adhesive with a broad approved temperature range (Polytec Polymere Technologien, EC 101,
Waldbronn, Germany) was applied to the joining surfaces of all wall and bottom parts immediately prior to bolting them together. In addition, after assembly, a silicon sealing (Dow Corning 3145) was applied to all inside seams of the enclosure. Inserted into the front plate and sealed with a silicone O-ring is a connector panel with one ½" bulkhead connector (Swagelok) for the sampling gas stream (see Section 2.4), four ¼" bulkhead connectors (Swagelok) for pressure release and flushing of the enclosure, two sealed USB (USB1/2 in Table 2) and RJ45 sockets (ETH1/2) and an SMA-RP connector
(WIFIA) to attach a WIFI antenna (see Section 2.5). In the bottom plate, another O-ring sealed connector panel holds two hermetically sealed connector sockets (P1 and P2) for electrical connection to the power box (see below). The pump (D8 in Table 2, cf. Section 2.4) is also bolted to the bottom plate, and another ½" bulkhead connector (Swagelok) is integrated next to it to connect to the pump on the outside and to the end of the sampling gas line on the inside.

*Design of the power box*

Base plate, walls and lid of the power box are made of 2.5 mm thick aluminium sheet metal (EN AW 5052 H111). It is mechanically attached to the enclosure at each corner and one near the centre by five M5x125 screws, each supported by a tubular bushing (5.3 mm inner diameter, 10 mm outer diameter, three made of stainless steel and two of carbon fibre, each with 4 mm wall thickness) to obtain sufficient pre-tensioning. Wiring between the power box and the enclosure is done by two connectors S1 and S2 that attach to the sockets P1 and P2 in the enclosure bottom plate through an opening in the power box'
lid. Another opening in the lid allows the pump to slide into the volume of the power box. Electrical components including AC-DC and DC-DC converters, EMI filters, temperature controllers and a data logger (all described in Section 2.5) are attached either to the bottom or to the side walls of the power box. Two connectors (J5 + J6) at the side walls of the power box provide the 24 V power to the external fans of the thermoelectric assemblies mentioned above. In the front of the power box, there are two connector panels, one holding the main AC power supply socket (J1) and miniaturised, aircraft style thermal



circuit breakers with push/pull on/off manual actuation (B1 – B5), and one holding three additional connectors (J2 – J4). The purpose and wiring of all connectors are described in Section 2.5. The power box is not pressure tight, but gaps in the housing are avoided for EMI considerations.

*FEA calculations*

An FEA-Model of the AMICA structure loaded with twelve different load cases was run to determine the mechanical
strength. All twelve load cases were restarted from the base load case that includes pre-tensioning and embedding of the screws at room temperature. For this base load case some local plasticity was found after pre-tensioning of the screws in the right-hand side power box sheet. The von Mises equivalent stresses are below the yield strength and henceforth below the ultimate strength after embedding of the screws.

Six operative load cases at flight level with accelerations set at ± 4 g in both flight direction and horizontal transversal to the
flight direction and ± 7 g in vertical transversal to the flight direction were analysed in conjunction with an ambient temperature of -60 °C and an internal excess pressure of 1000 hPa in the enclosure. At these expected operative loads, repeated occurrence of plasticity in parts should be avoided as much as possible to prevent a low-cycle fatigue failure, and the results indicate that this is fulfilled. No additional local plasticity was found for the operative load cases in the right-hand side power box sheet, but some local plasticity was found in the top power box sheet. The von Mises equivalent stresses are below or
at the yield strength at these locations, but far away from the ultimate strength. Existing plasticity will not increase further in a successive flight.

Six emergency load cases with accelerations set at 10 g in all directions were chosen to simulate two situations. The first is the transport of AMICA through airfreight, while the second concerns acceleration specifications given by the operators of the carrier airplanes and valid for emergency landing conditions. Since both situations have the same environmental conditions
tions with respect to ambient temperature and internal excess pressure, they are considered as one situation. The accelerations applied are valid for airfreight with a safety margin of +10 %, and encompass the accelerations occurring at emergency landing conditions. At this special load level plasticity in parts is allowed, but failure leading to disintegration of parts is prohibited. Some additional local plasticity was found for the emergency load cases in the right-hand side power box sheet and at one of the bore holes of the rear grey frame plate. The von Mises equivalent stresses are slightly below or above the yield
strength at these locations, but far away from the ultimate strength.

*Aircraft specific mounting considerations*

For HALO operation, AMICA is mounted in a standard rack (R-G550SM, EPA-DLR-00004-000) using a set of adapters (for details, see supplementary Figure S1). It conforms to all requirements with respect to total weight and position of the centre of gravity, and thus the mechanical airworthiness certification is inherited from that of the rack. Because the rack is mounted
inside the cabin with a set of preinstalled shock absorbers, no additional vibrational isolation hardware is used.



On M55 Geophysica, AMICA is installed inside a dome on top of the aircraft. Specific mounting plates have been designed to attach AMICA onto the base frame of the dome (see supplementary Figure S2), including four springs (Enidine WR12-300-08) with the following characteristics: in normal direction a maximum force per spring of 4.65 kN resulting in a spring deflection of 37.1 mm and in both shear directions a maximum force per spring of 5.55 kN resulting in a spring deflection of

39.1 mm. The springs are designed to withstand the normal and shear forces that occur due to the different load cases. For the operative and emergency landing load cases the springs will stay elastic. For airfreight emergency loading and then only if the stowage of AMICA occurs perpendicular to the prescribed flight direction, the two front springs will exceed their elastic bearing capacity and will hit the internal limit stop. Another purpose of the springs is to decouple the instrument from the aircraft body movements, mainly to absorb potentially heavy shocks during take-off and landing. The effectiveness of the

springs was tested during the first deployment using two vibration sensors (SlamSticks, Mide Technology LOG000200-0006, Medford, Massachusetts) attached to each side of one spring. The vibrational data of several hours was cut in time sequences of 30 seconds of data each. Then from every time sequence a fast Fourier transform (FFT, in the range from 0 to 1000 Hz) and subsequently a power spectral density (PSD) and a cumulative power spectral density (CPSD) was made to obtain the root mean square (RMS) values of the directional accelerations ($g_{RMS}$), both for the data on the M55 side and on

the AMICA side of the springs. The attenuation of the vibrations was then calculated from the ratio of the $g_{RMS}$-values. Depending on the direction the attenuation lies between -18 dB for the flight direction and -25 dB for both directions perpendicular to the flight direction (for more detail, see Supplementary Figure S3).

For mounting, AMICA is lifted onto the aircraft by crane and the exact position is adjusted by hand. For this purpose, four shackles and hand bars are attached to the enclosure at the four corners (also included in Figure S2).

Because AMICA is mounted to the M55 Geophysica as a unique entity, mechanical stability had to be certified and documented. In its Geophysica setup, AMICA is laid out for elastic deformation up to 7 g with fully preserved functionality and plastic deformation up to 10 g. This was simulated in the FEA calculations described above. In addition, a shaker test with a dummy of the AMICA housing internally equipped with dummy weights closely resembling the distribution of the electronics and ICOS hardware was carried out (at MOOG, CSA Engineering, Mountain View, California) according to the test pro-

cedure RTCA/DO-160G (elastic deformation for 7 g acceleration in X, Y and Z direction) and successfully passed. It confirmed that the AMICA housing responded to a 0.5 g sine sweep, before and after the application of random vibrations with no significant difference in system behaviour.

**2.4 Sampling and flow system**

The sampling system has been designed to ensure rapid transfer from the inlet to and through the cavities (effective instru-

ment time resolution is ultimately limited by the cavity flush time) and to keep pressure inside each cavity constant to warrant straightforward analysis of the ICOS spectra with good precision. Cavity pressure $P_{cav}$ is chosen as a compromise between absolute sensitivity (pressure correlates with number density and therefore absorption of each species) and spectral

resolution (which deteriorates at higher pressure as a result of pressure broadening). Depending on the expected mixing rati-
os of the measured species and the wavelength separation of their absorption peaks, pressures employed in ICOS systems
typically range from a few hPa to about 200 hPa. Because AMICA is operated on aircraft, the lowest possible cavity pressure
is further limited by the ambient pressure at the sampling inlet.

*General setup*

A schematic of the AMICA sampling and flow system is shown in Figure 3. Sampling air enters the system at a ½" bulkhead
connector port (Swagelok) in the enclosure wall, equipped on the outside with a 7 µm filter (Swagelok SS-4FW7-7) coated
with Sulfinert® (SilcoTek GmbH, Bad Homburg, Germany) to prevent dust from entering the system. Inside the enclosure,
the air flows through a system of tubing (Sulfinert® treated 3/8" and ¼" stainless steel tubing), valves (C5, see below) and
the two cavities in series. A second Sulfinert® treated filter with 2 µm pore size (Swagelok SS-4FW4-2) is placed directly
upstream of the first cavity to prevent small particles that have passed through the first filter or released from the valve seals
to enter the cavities and contaminate or damage the mirrors. The two 508 mm long cavities each have a volume $V_{cav}$ of 0.911
L and are coupled in series.

The sampling air is drawn into and through the entire system at flow rates $F$ between 0.8 SLM (with $P_{cav} \sim 45$ hPa) and 1.6
SLM (with $P_{cav} \sim 80$ hPa) by a pump (D8) placed downstream of the second cavity and a check valve to avoid back flow into
the system. The pump exhaust blows air directly into the power box for extra ventilation therein. The Vaccuubrand MD1
Vario pump was selected as a compromise between weight, power draw, internal heat generation and flow rates at typical
AMICA cavity pressures.

*Pressure regulation*

Because the ambient pressure can range from ~ 1000 hPa at ground level down to about 55 hPa at the highest flight altitudes
of 20 km, a system of two parallel proportional solenoid valves with orifices of 0.762 mm (C5a) and 3.2 mm (C5b) is used to
precisely regulate cavity pressure $P_{cav}$. The valves are controlled by separate pressure controllers (C3a/b) with the set point
of controller C3a being ~ 1 hPa smaller than that of controller C3b. There is a pressure gauge (C4) at each cavity, but only
the reading from gauge C4a at cavity 1 is wired to the controllers (C3a/b) for pressure regulation.

Pressure regulation and response of $P_{cav}$ to ambient pressure have been tested in the laboratory by pumping down a 50 L bot-
tle through the original AMICA inlet tubing used in the HALO aircraft (cf. below) with two additional pressure gauges
placed at the bottle (Figure 4). At ambient pressures above ~ 200 hPa, cavity 1 pressure is typically regulated to ± 0.2 hPa
around the higher set point, with the larger valve remaining fully closed because the lower set point is not reached. When
ambient pressure drops below ~ 200 hPa, the resistance of the smaller valve C5a becomes too large even when fully opened,
and the cavity pressure starts to drop below the set point of the corresponding regulator C3a. When this happens, the second
regulator C3b with the slightly lower set point starts to open the larger valve C5b, which allows for pressure regulation to ±

0.6 hPa down to an ambient pressure about 10 hPa larger than the lower set point. When ambient pressure drops further, both
valves remain fully open, and the cavity pressure will drop and vary with ambient pressure at a few hPa below it. As a consequence of the additional flow resistance of the tubing between the cavities, the pressure inside the second cavity is approximately 1.5 hPa lower than the cavity 1 pressure.

*Aircraft specific inlets*

During aircraft operation, sampling air is taken in through a primary intake sticking out of the aircraft boundary layer, and
then needs to be transferred to the instrument inlet. For AMICA, this has so far been implemented for the in-cabin operation
on the German HALO aircraft, and for the operation in a dome on top of the high altitude aircraft M55 Geophysica. Both
inlet systems are briefly described and characterized here.

On HALO, a rear facing 0.5 inch stainless steel tube in a standard Trace Gas Inlet (TGI) near the front of the aircraft is used
as primary inlet. The diameter is reduced to 3/8" at the inlet, and the air is transferred to the filter right in front of the AMI-
CA instrument inlet via a 214 cm long ¼" inner diameter Sulfinert® (SilcoTek, Bellefonte, USA) coated stainless steel tube
with a 10 cm Sulfinert® coated bellow on each side to avoid stress on the connectors. The primary inlet is not actively heated
at the tube used for AMICA, but heat transfer from a neighbouring inlet tube ensures it to be warmer than ambient temperature. The transfer tube inside the cabin is neither heated nor insulated.

A dedicated shared primary inlet with three separate rear-facing tubes for AMICA and two other instruments was developed
for the dome on top of the M55 Geophysica. The AMICA tube is a 40 cm long 3/8"sulfinert coated stainless steel tube. A
200 cm long ¼" inner diameter Sulfinert® coated stainless steel tube transfers the air from the primary inlet to the instrument.
As described for above for HALO, two bellows are placed at each side of the transfer tube to avoid stress and breakage.

The time lag for the sampling air to flow from the inlet outside the aircraft to cavity 1 is approximately 6 seconds at ground
level and 0.6 seconds at an ambient pressure of 100 hPa, corresponding to a distance of 120 m at an aircraft speed of 200
m/s. The additional time lag for the second cavity is 2.8 seconds (equivalent to 560 m at an aircraft speed of 200 m/s). The
flush time for each cavity (given by $V_{cav} \cdot [P_{cav}/1013 \text{ hPa}] \cdot [273 \text{ K}/T_{cav}]/F$) is about 2.5 seconds, which sets a limit to the actually useful time resolution of the measurement to 0.4 Hz (equivalent to 500 m distance at an aircraft speed of 200 m/s).

## 2.5 Power concept and electronic design

*AC power supply*

A complete block diagram of the power box is shown in Figure 5. AMICA is powered only by a single phase AC power
supply line through connector J1 at the power box connector panel. Inside the power box, the AC current is passed through
two filter modules F1 and F2 to minimize conducted EMI interaction between AMICA and the aircraft power system. It is
then distributed to supply power to four component groups: (i) an inlet heating, (ii) pump and fans, (iii) enclosure tempera-

ture control system, and (iv) ICOS measurement system. Prior to any components or power converters, the current is passed through thermal circuit breakers (B1 – B5) to protect the system from currents exceeding their nominal values (e.g. due to component malfunctions or unforeseen short circuits). The breakers B1, B3, B4 and B5 are chosen with current limits of 2.5 A, 2 A, 6 A and 4 A for component groups (i) to (iv) respectively, reflecting their expected maximum power consumption during nominal operation. Additionally, the breakers allow for push/pull manual actuation to switch on/off power to individual component groups separately for test and diagnostic purposes.

*Inlet heating*

Inlet heater elements (i) are directly powered by the AC current. A temperature controller D10 in combination with a relay D9 allows the control of the inlet temperature to a set point (set in the D10 menu). Power to the inlet and a PT100 temperature probe for control are wired through connector J3. Inlet heating has been used with a set point of 30 °C during M55 Geophysica deployment, but was disabled during operation in the HALO cabin.

*AC-DC conversion*

All other components in AMICA require DC power. This is generated by AC-DC converter modules V1 – V3. These converters from Vicorpower have autoranging AC input that automatically sense the AC supply voltage between 90 and 132 V as well as 180 and 264 V, and the frequency over a range of 47 – 440 Hz, so that AMICA can be operated inside a laboratory (115 or 230 V @ 50 or 60 Hz) and inside aircraft (typically 115 V @ 400 Hz) without any internal modifications. The output

of V1 – V3 is 24 V DC each. Self-made add-on boards are installed directly onto the output terminals of V1 – V3 to sense the currents for monitoring by using special Hall ICs. The boards also detect the output voltage and distribute the output power into several current limiting paths by using small SMD fuses specially designed for space limited circuit boards ("nano fuses", Littelfuse, Chicago, USA).

*Pump*

V1 powers the pump D8 through connector S3 and the two fans attached externally to the TEC assemblies through connectors J5 and J6. Pump and fan power lines are fused at 6 A and 0.5 A each respectively.

*Temperature regulation of the enclosure*

The TEC assemblies themselves are independently run by two TEC controllers (D7a/b) powered by the two V2 output channels, each fused at 10 A. Each TEC assembly consists of 16 Peltier elements (30x30 mm$^2$). Both sign and magnitude of volt-

age and current are modulated by the TEC controllers (D7a/b) in order to stabilize the measured temperature as near as possible to the 35 °C set point (set in the D7 menu via PC interface using a Mini USB service port). The NTC (1M) temperature probes are placed inside the enclosure at some distance from the TEC assemblies but each closer to the TEC assembly connected to the same controller as the probe (exact positions are shown in Figure 2). Cables for power wires and temperature probes are run from the controllers into the enclosure via the S2/P2 connection.



*Power supply to ICOS components*

All components of the actual ICOS measurement system are powered by V3 through two output channels fused at 10 A each. Channel 1 provides power to power box temperature board D4, the laser controllers C6 and components C3, C4, C10 and C12 that operate at 12 VDC. The 24 VDC supplying the laser controllers C6a/b is passed through a microRAM (F9) for EMI filtering and subsequently fused at 4 A before being passed through connection S1/P1directly to the LTC-1141 boards

C6a and C6b, which handle powering of the respective lasers and their TECs. A parallel line from V3 Channel 1 is passed through EMI filter F6, from which separate lines lead to temperature board D4 and DC-DC converter D3. D4 provides 5 VDC power to temperature sensors inside the power box (see Housekeeping below) and ± 12 V to the cavity temperature sensors in the enclosure through connection S2/P2. D3 provides 12 VDC and two output lines, each fused at 6.3 A, pass the voltage via the S1/P1 connection to two distribution boards inside the enclosure that supply the 12 VDC components in their

vicinity (see Table 2 for the detailed allocation). V3 Channel 2 is first passed though EMI filter F5 and then divided into three supply lines. One of these, fused at 4 A, transfers 24 VDC directly through S1/P1 to distribution board C13b, from which all fans (listed in Table 2) in the ICOS enclosure receive their power. The second filtered line from V3 Channel 2 is converted to 5 VDC in D1, fused at 10 A, and passed through S1/P1 to distribution board C13a. The 5 VDC power the embedded PC (C1) and SSD (C2) as well as two sensors placed on C13a that monitor temperature (Texas Instruments, LMT86)

and pressure (Infineon, KP215F1701) inside the ICOS enclosure. The third filtered V3 Channel 2 line is used as input for DC/DC conversion board D2 to generate two independent ±12 VDC sources with particularly low ripple output to provide up to 0.5 A to the two highly sensitive pre-amplifiers C9a and C9b through the S1/P1 connection.

*Grounding*

The AMICA grounding concept distinguishes between three grounding systems: earth ground (chassis ground), supply

grounds, and analog/digital grounds.

Earth ground plays a special role as AMICA operates on high AC voltage (100 to 250 VAC at up to 12 A). Due to the two EMI filters F1 and F2, which utilize Y capacitors between the live and neutral conductor, leakage current occurs and flows from the live conductor to the filters' casings, which are connected to the power box metallic housing, and flows back to the power source. This leakage current may flow through other paths (such as a human body touching the instrument) and can

cause electric shock, if the ground is inefficient or interrupted. As the entire AMICA housing (ICOS enclosure and power box) is electrically conductive, the whole chassis becomes an earth ground and must be grounded to the power source properly. To ensure that AMICA is always well grounded to system power during lab operation, the dedicated grounding threaded rod at the power box front panel has to be connected to the ground using an earthing strap. For operation on aircraft, additional grounding interfaces are placed at the four corners of the enclosure. Internal electric components benefit from the

instrument housing's good EMI characteristics as it keeps out external disturbances of any kind and vice versa.





Inside the instrument, supply grounds and analog/digital grounds are managed to connect to ground according to the internal assemblies and structures of the instrument with the goal of reducing problematic internal EMI and minimizing noise coupling between components. Interconnections are laid out to avoid potential internal ground loops wherever possible. The other measure is to provide electrical components or component groups with independent and isolated power sources. This is

done for example by DC/DC conversion board D2 to supply the two high sensitive pre-amplifiers C9a/b without ground loops.

*Housekeeping*

As mentioned above, voltages and currents of each AC/CD converter output channel are monitored by using self-made PCBs as add-on boards for V1 – V3. For each voltage monitor, a high precision DC voltage isolation sensor (ACPL-C87AT,

Broadcom) is used that utilizes optical coupling technology with a fully differential amplifier to provide an isolated analog output signal. Current sensor ICs (ACPL-C87AT, Allegro MicroSystems) each use an integrated Hall transducer to measure the magnetic field of the applied current flow and convert it proportionally into an isolated voltage. These components were chosen following the grounding concept to reduce the signal ground loops. Temperatures of the three AC/CD converter boards and the pump are monitored using integrated circuit temperature sensors (Texas Instruments, LMT86) powered and

read from temperature board D4.

Voltage, current and temperature signals are digitized by a LabJack T7 data logger powered from the embedded PC (C1) via a USB line passed through connection S2/P2. Through the same USB line, the set of 14 parameters is regularly read by the software on the embedded PC. The monitoring of these parameters is needed to observe smooth operation of each component in the power supply system, and the data can help in case trouble-shooting becomes necessary.

In the enclosure, signals from the temperature and pressure sensors on the C13a board and from the temperature probes C14 and pressure gauges C4 of the two cavities are acquired by the embedded PC from analog input channels of a data acquisition card (RedWave S310) in the embedded PC/104 stack. Parameters related to the lasers C7 (laser and heat sink temperatures) and their driver boards C6 (voltage, board and processor temperature) are communicated via UDP data stream together with the spectra from the C6 boards (see *Communications* below and Section 3.1).

*Communications*

AMICA contains a self-contained and fully operational embedded PC (C1; note that a previous version of AMICA contained two independent embedded PC/104 stacks, cf. Section 5) equipped with a 64 GB SATA SSD (C2) and running under a Linux operating system (Lubuntu 18.04). Communications ports are wired to appropriate connectors at the enclosure connector panel (details given in Table 2). Two USB ports (USB1/2) allow connection of computer periphery such as keyboard

and mouse as well as the use of USB memory devices. An RJ45 connector (ETH1) allows for LAN connections to a laptop, PC or an external network via the PCs 1 Gbit Ethernet port. However, cable based LAN connections are not always feasible,

e.g. on the M55 Geophysica when the dome cowling is closed and the instrument can't be accessed. To still be able to communicate with AMICA, a wireless LAN module (RTD, WLAN18202ER) in the PC/104 stack is wired to an SMA connector (WIFIA) in the enclosure connector panel with a WIFI antenna attached on the outside. With this setup, WIFI connections

between AMICA and a laptop or desktop PC are possible up to about 200 m distance. Another RJ45 connector (ETH2) is used for VGA extension via RJ45 cable that allows the connection of an external screen to the PC's VGA port even when the enclosure is fully closed. This can be useful to directly work on the AMICA PC in the laboratory, or for trouble shooting if an external Ethernet connection can't be established.

A PC/104 module with two additional Gbit Ethernet channels (RTD, LAN18222HR) is used for internal communication

between the embedded PC C1 and the LTC-1141l boards C6a/b. A TCP/IP protocol is used for initialization of the boards at software start-up, and for sending commands (e.g. to change the temperature set point) to or receiving status information from C6a and C6b. A data stream consisting of the acquired and averaged spectra and some additional parameters (cf. Section 3.1) is transferred one-way from each LTC-1141 to the PC via UDP data stream.

*EMC test*

To certify proof of airworthiness, an electromagnetic compatibility (EMC) test was carried out according to the environmental conditions and test procedures for airborne equipment (RTC-DO160 E, category M; testing was done at steep GmbH, EMC Center Bonn, Germany). AMICA passed the test procedure without exceeding any of the given thresholds.

## 3 Data acquisition and analysis

Instrument control and data acquisition are handled by a software package written in PYTHON 3.7.6 that runs under a Linux

environment (Lubuntu 18.04) on the PC/104 stack in the pressurized enclosure. Below, we briefly describe the implementation of data acquisition and storage and the algorithms used to analyse the ICOS spectra and retrieve trace gas mixing ratios.

### 3.1 Data acquisition

In the current version, the AMICA software uses one continuously running data acquisition loop to consecutively read spectra and housekeeping data (a complete list of monitored parameters is given in Table 4):

– **Time averaged ICOS spectra from each LTC**: at each time interval $t_{int}$, the LTC sends an ICOS spectrum averaged from $N_{ramps}$ (cf. Section 2.2) together with 6 parameters related to laser and LTC status (see Table 4 for details) to the PC via UDP stream in packages of 256 floating point numbers each. The packages are read by a socket command inside a loop that puts the housekeeping parameters into corresponding named variables and the spectra into a netCDF type structure, which is synchronized to the SSD hard disk every $60^{th}$ data acquisition cycle.





–   **Cavity and enclosure pressures and temperatures**: both pressure and temperature sensors in the enclosure and in each cavity are sensed as analog voltages and converted to digital numbers by the RedWave S310 data acquisition card in the PC/104 stack. The six parameters (listed in Table 4) are read into named variables in the AMICA software via the S310 application programming interface (API).

     –   **Power box temperatures, voltages and currents**: the AMICA software calls the LabJack T7 (C6) to transfer the

readings on the 14 AD channels (given in Table 4) via USB connection. Actual temperatures, voltages and currents are calculated from the received voltage signals and stored in named variables.

At the end of each data acquisition cycle, all housekeeping parameters listed in Table 4 are converted into a single string and added to an ASCII file on the SSD. During HALO flight missions, a selection of these data (marked in Table 4) is additionally sent to the aircraft server as a UDP stream. An interactive software version for laboratory use during tests and calibrations

displays all housekeeping data in a widget displayed on the screen (a screenshot of this widget is given in the supplementary Figure S4).

### 3.2 Analysis of ICOS spectra

Three different sections are extracted from the averaged ramps (illustrated in Figure 6) and analysed. First, the points between the start and the laser turn on (grey region in Figure 6) are averaged to calculate the detector dark signal $S_{dark}$. Second,

an exponential decay is fitted to points (the broad region where this fit is made is marked red in Figure 6) between the laser turn off and the point when the signal has decayed back to $S_{dark} + 3\sigma (S_{dark})$ to deduce the ring down time $\tau$ (using the method of linearly fitting the time shifted signal given in Sayres et al., 2009). To avoid contribution to the fit of the actual decay time of the laser output and the time for the laser light to be coupled into the cavity, points within approximately $2\tau$ after laser turn off are excluded from the fit. Because the background $\tau$ of an empty cavity is needed to determine mirror reflectivity $R$

and the mean absorption free path length $L_0$, it is important that the laser ramp ends in a spectral region without any absorption by trace gases present in the atmosphere. From the fitted $\tau$, $L_0 = c \cdot \tau$ and $R = (1 - L_{cav} / L_0)$ are calculated with the speed of light $c = 2.998 \times 10^8$ m s$^{-1}$ and cavity length $L_{cav} = 508$ mm.

Third, the part of the active laser ramp (blue region in Figure 6) is extracted over which spectral fitting is employed to deduce trace gas mixing ratios. Prior to spectral fitting, the points/time along the ramp have to be converted to a wavenumber

scale. For each laser source, an etalon fit was experimentally determined when the temperature and ramp current settings were initially tuned to the desired wavenumber range. In Equation (1), the wavenumber is expressed as a function of laser current:

$$\nu(I_{laser}) = \nu_0 + 1.0 \cdot 10^9 [a_1 + a_2 \cdot I_{laser} + a_3 \cdot I_{laser} / \log_e (I_{laser}) + a_4 \cdot \log_e (I_{laser}) / I_{laser} + a_5 / I_{laser}] / c \qquad (1)$$

Here, $\nu_0$ is a fixed wavenumber within the spectrum (typically chosen at or in relation to prominent absorption peak within

the spectral range) and the bracketed part is the shift in GHz from this peak with fitting parameters $a_1 - a_5$, determined using





an etalon. For this equation to work, the $\nu_0$ point in the spectrum needs to remain in the same position in terms of $I_{laser}$, which is usually the case after temperatures of the laser and the laser housing have stabilized. Small drifts are actively adjusted by the software adding an offset (< 0.1 K) to the temperature set in the LTC, and the wavenumber to $I_{laser}$ relation given by Equation (1) holds with the same fitting parameters when this is done. This is regularly checked and validated by testing if
all peaks appear in the correct positions when measuring standards with many absorption peaks present in the spectral range. Using a relation with respect to $I_{laser}$ rather than points or time along the ramp in Equation (1) allows for the possibility of applying a curved ramp of $I_{laser}$ resulting in a quasi-linear wavenumber scale that makes spectral fitting easier.

Spectral fitting following the method described by Sayres et al. (2009) has been implemented in IDL code and has been used to process the spectra recorded during the M55 Geophysica field campaigns described in Section 5 (note that proprietary
ABB-Los Gatos Research Software was used for instrument control and spectra acquisition during these campaigns, with two separate PCs handling the two ICOS channels). Encoding of the full spectral fitting algorithm in Python is currently in progress and shall not only be used for post-processing of recorded spectra but also for real-time processing during measurement flights. In the current Python software actually running on the AMICA instrument, a less computationally demanding approach is used to fit trace gas concentrations: using an experimentally measured baseline (with the cavity filled with
Argon at 99.9999% purity), the spectrum is transformed into absorption space and a Lambert-Beer fit of wavelength dependent absorption $A(\lambda)$ is solved for the concentrations $c_i$ of different absorbers with

$$A(\lambda) = L_0 \sum_i c_i \alpha_i (\lambda) \qquad (2)$$

where $\alpha_i(\lambda)$ is the wavelength dependent absorption coefficient for each absorber based on the line parameters from the HITRAN2012 database (Rothman et al., 2013). This method is only an approximation as it does not take into account the cavi-
ty broadening effects (cf. Sayres et al., 2009), which are small as long as

$$L_{cav} \, c_i \, \alpha_i(\lambda) \ll 1 - R \qquad (3).$$

When this condition is not met, the effective path length $L_{eff}$ is reduced compared to $L_0$ determined for the empty cavity and the sensitivity is reduced and the analysis is complicated because the variation of $L_{eff}$ with wavelength over a broadened absorption peak leads to additional broadening in the observed spectrum compared to the Voigt profile calculated from the
broadening coefficients in HITRAN.

**4 Characterisation of implemented ICOS measurement setups: spectra and calibrations**

In this section, different laser, mirror and detector setups to measure specific target gases are described and characterized. An overview of these setups is given in Table 2, detailed descriptions are given in the subsections below. Beside these, a wide variety of setups is theoretically possible, targeting many trace gases.



### 4.1 OCS, $CO_2$, CO and $H_2O$ at 2050 – 2051 cm$^{-1}$


A quantum cascade laser (QCL, Hamamatsu), operated at 18.6 °C and ramped over the current range 268 – 360 mA, emits light over the wavenumber range of 2050.23 – 2051.47 cm$^{-1}$. In this spectral window, OCS (major absorption bands at 2050.39 and 2050.86 cm$^{-1}$), CO (major absorption band at 2050.80 cm$^{-1}$), $CO_2$ (major absorption band at 2050.60 cm$^{-1}$) and $H_2O$ (major absorption bands at 2050.63 cm$^{-1}$) can be measured. Several $O_3$ absorption bands also exist in this window, but

absorption at typical cavity pressures is completely negligible even at stratospheric ozone levels in the ppm range.

A set of typical spectra in both intensity and absorption domains for this setup is shown in Figure 6. Absorption peaks for $CO_2$, $H_2O$ and OCS are spectrally well resolved, while the CO peak at 2050.80 cm$^{-1}$ significantly overlaps with the OCS peak at 2050.86 cm$^{-1}$. As both intensity domain and absorption domain spectral fitting (see Section 3.2) are done with full forward simulation of spectra derived from HITRAN line parameters, the overlap of these peaks is not critical and does not

introduce any bias over the range of typical atmospheric concentrations (cf. calibrations below). The full spectral fitting also compensates for $H_2O$ absorption near the OCS peak at 2050.39 cm$^{-1}$ becoming significant at high $H_2O$ mixing ratios up to 3 % encountered in the tropical troposphere. Because OCS absorbs only weakly at typical atmospheric mixing ratios around 500 ppt in the troposphere and often significantly lower in the stratosphere, a large $L_{\text{eff}}$ exceeding about 1000 m is needed to measure it with adequate precision. The high $R$ required to achieve such high $L_{\text{eff}}$ introduces difficulties in the analysis of

signals for strong absorbers such as $CO_2$ at atmospheric concentrations, because the condition given in Equation (3) is not met and $L_{\text{eff}}$ is significantly reduced at the $CO_2$ absorption band and it varies over the broadened peak. This reduces sensitivity and introduces additional broadening (more details on this are given in Figure S5 in the supplementary material).

Results of laboratory calibrations against series of known standards (a compilation of standards and standard preparation procedures used in this work is given in Table 5) are shown in Figure 7 for OCS and in Figure 8 for CO. For both gases, ex-

cellent agreement with the standards is achieved just using the HITRAN parameters for the fit, i.e. no calibration factors are used in the derivation of their mixing ratios from the spectra. Taking into account uncertainties in the standard and in the spectral analysis procedure, accuracies better than 5 % are estimated for both OCS and CO. Due to uncertainties in the prepared standards at low CO mixing ratios (caused by both a large dilution ratio and the possible presence of residual CO of unknown concentration in the clean dilution gas), accuracy is estimated to be only better than 15 % at CO mixing ratios be-

low 60 ppb. Figure 7 also shows good agreement between measured and standard OCS concentrations during calibrations carried out at reduced pressure (simulating the high altitude range during the M55 Geophysica campaigns, suggesting that low cavity pressure does not cause systematic errors. It must be noted, however, that at such low pressures, precision significantly deteriorates (cf. Section 2.3) and low stratospheric OCS and CO mixing ratios cannot be detected anymore.

For $CO_2$, spectral fits in either intensity or absorption space using only HITRAN parameters fail to arrive at the mixing ratios

of the standards used or those expected in atmospheric measurements. This is at least partly caused by the issue with the strong $CO_2$ absorption affecting the absorption path length, but we also note that unlike the HITRAN parameters used for





OCS and CO, the parameters for the $CO_2$ line used are not based on experimental data but on theoretical calculations and are therefore associated with higher uncertainties. As a result, an additional calibration factor, determined from calibrations against known standards, is introduced when deriving $CO_2$ mixing ratios from spectral fits.

Precision for OCS, CO and $CO_2$ is shown in the form of Allen deviation plots determined from measuring one standard over a longer period (Figure 9). For all three gases, precision can be improved at the cost of time resolution by averaging up to one or two minutes where the curves reach more or less pronounced minima. Over the time periods tested it is also seen that the σ does not significantly rise again at longer times. We expect to achieve a further lowering of the σ curves in the future by further reducing electrical noise.

In the current setup, $H_2O$ can only be detected at mixing ratios above 100 ppm. Comparison of mixing ratios derived from spectral fits based on HITRAN parameters with a relative humidity sensor (MSR 165) placed near the AMICA inlet showed agreement better than 4 % under different conditions (4000 – 30000 ppm $H_2O$).

Because OCS is one of the main scientific interests of our research group, the 2050 – 2051 $cm^{-1}$ setup is typically used in AMICA in the cavity 1 position.

**4.2 $O_3$ and $NH_3$ at 1033 – 1034 $cm^{-1}$**

A quantum cascade laser (QCL, Hamamatsu), operated at 29.6 °C and ramped over the current range 775 – 925 mA, emits light over the wavenumber range of 1033.21 – 1034.36 $cm^{-1}$. In this spectral window, $O_3$ (major absorption bands at 1033.24, 1033.35, 1033.68, 1033.86 and 1033.93 $cm^{-1}$) and $NH_3$ (major absorption bands at 1033.32 and 1034.01 $cm^{-1}$) can be measured. A spectrum measured by this channel during the 2019 SouthTRAC deployment (Section 5.3) is shown in Figure 10

together with a spectral fit that uses HITRAN line parameters.

Laboratory tests were carried out with an $O_3$ generator that accurately measures $O_3$ concentrations with an internal UV absorption spectrometer (Proffit and McLaughlin, 1983). As shown in Figure 11, AMICA underestimates $O_3$ mixing ratios. The reason for this underestimation has not yet been fully understood. One possible reason could be loss of $O_3$ inside the transfer line or the AMICA instrument, but we deem it unlikely for this to be the only reason. Inside AMICA, all surfaces are

either Sulfinert® or Teflon except for small stainless steel parts inside the valves, and removal of a metal connector from the transfer line had no significant effect. We also note that for any particular mixing ratio, the AMICA response remains reasonably constant, and there we observed no significant difference going from low to high concentrations and vice versa. Another reason could be inaccuracies in the wavenumber scale for this channel, which is locked to a significantly less prominent absorption feature than the $CO_2$ peak in the 2050 – 2051 $cm^{-1}$ channel. Finally, some fraction of the bias may be related

to uncertainties in the HITRAN line parameters used.

In relative terms, the underestimation decreases with increasing mixing ratios, ranging from about 30 % at 100 ppb to 15 % at 800 ppb, but the response is linear overall, with a negative intercept of about 22 ppb (Figure 11). Further tests and calibra-





tions to better understand the reasons for the underestimation and to see if the response function is constant over longer periods of time and can thus be used as a correction function are planned.

Significant absorption at the $NH_3$ line positions has not been detected during the SouthTRAC flights. Tests with a laboratory standard to investigate the sensitivity of this measurement are planned.

### 4.3 $N_2O$, HCN and $C_2H_2$ at 3331 – 3332 cm$^{-1}$

An interband cascade laser (ICL, nanoplus), operated at 16.5 °C and ramped over the current range 50.0 – 67.5 mA, emits light over the wavenumber range of 3330.6 – 3332.2 cm$^{-1}$. In this spectral window, $N_2O$ (major absorption band at 3331.65

cm$^{-1}$), HCN (major absorption band at 3331.58 cm$^{-1}$) and $C_2H_2$ (major absorption band at 3331.33 cm$^{-1}$) can be measured.

Figure 12 shows spectra recorded with this channel for diluted $C_2H_2$ from a gas bottle and on the ground and in-flight during the 2017 deployment from Kathmandu (Section 5.2). The pattern of the $C_2H_2$ peaks in the spectra from the gas bottle and of the $H_2O$ peaks in the spectra taken at ground level in Kathmandu show that the OA-ICOS measurement in the desired wavelength region was successfully implemented. However, the decay of light intensity at the end of the ramp was too fast to ob-

tain a useful ring down fit. Based on the absorption obtained for the $C_2H_2$ mixing ratios used, mirror reflectivity is estimated to be only ~ 0.995, corresponding to an effective path length of about 100 m. This is not sufficient to measure $N_2O$, HCN and $C_2H_2$ absorption at atmospheric concentrations, which is confirmed by the absence of absorption features in the spectra recorded at high altitude during the 2017 deployment in Nepal (Section 5.2). An mirror reflectivity of at least 0.9995 and probably a higher output ICL are needed to implement such atmospheric measurements, and we hope to test this with AMI-

CA in the near future.

### 5 In-flight performance

To date, AMICA has been operated in the field during three aircraft campaigns. Between them, the instrument has undergone major technical modifications to continuously improve in-flight performance. Because a full description of each individual setup with referencing of all individual components would render this paper extremely lengthy, the technical descrip-

tion in Sections 2 and 3 is limited to the current optimized version. To convey a broad sense for the "learning curve" during this instrument development phase and as a reference for scientists working with the AMICA campaign data (most of which is/will be freely accessible) or with published scientific results (e.g. von Hobe et al., 2020), reference to set ups or used parts different from the description in Sections 2 and 3 will be given for each campaign.



### 5.1 M55 Geophysica, Kalamata, 2016

The very first aircraft deployment of AMICA took place in Kalamata, Greece, with three flights in late August/early September 2016. A major aspect of this campaign, which was part of the European StratoClim project (Stroh and al., 2020), was to test several new or modified instruments in the M55 Geophysica payload.

At the time, two separate PC/104 stacks were used, one for each of the two ICOS systems. Acquisition and A/D conversion of the voltage signals from C9a/b was handled by a board inside each PC/104 stack, and averaging was done by the software
running on the PC. The data acquisition boards were also used to generate voltage ramps that were passed on to the laser controllers used at the time (RedWave Labs) for current modulation. Temperature regulation of lasers, laser mounts and detectors was by done by separate TEC controllers (Wavelength).

AMICA operated and produced useable data during each flight, but the data quality – precision in particular – was significantly reduced compared to the current instrument (i.e. as described in Sections 2 and 3) mainly for two reasons:

i) Cavity pressure fell below the set point of controller C3 b (42.5 hPa during this campaign) at an altitude around 14 km and was significantly lower than shown in Figure 4 for any given ambient pressure. This was largely caused by the flow resistance of two particle filters (Swagelok F series), which was remedied by replacing the filters by high flow versions (Swagelok FW series) before the third flight, in which cavity pressure regulation worked nominally up to about 16.5 km altitude. Above this altitude, the pressure drop between measured ambient and cavity pressure was still significantly larg-
er than in the laboratory experiment (described in Section 2.4 and shown in Figure 4). The likely reason for this is a significantly negative pressure coefficient at the dome on top of the M55 Geophysica aircraft during flight leading to an effective surface pressure at the inlet position up to 30 hPa below static pressure. This is supported by the observations of similarly lower than expected pressures in another instrument connected to an inlet directly adjacent to the AMICA inlet, and by a significant variability of the pressures in both instruments with the aircraft's angle of attack that is known to af-
fect pressure coefficients at the aircraft surface from fluid dynamics considerations. As stated in Section 2.3, lower cavity pressure reduces the trace gas absorption signals and therefore instrument precision.

ii) The second precision limiting factor is noise. The sophisticated clean power and grounding concept described in Section 2.5 had not been fully implemented in 2016, and ground loops as well as devices with a variable power draw (e.g. valves, fans, TEC controllers) connected to the same DC source induced noise at the data acquisition card in the two PC/104
stacks used at the time.

Another critical issue during this campaign was a tendency to overheating of the Power Box during the instrument warm up phase on the apron prior to take-off. When AMICA was turned on, temperature at the three VIPAC converters (V1 – V3) and near the centre of the Power Box rose above 80 °C within minutes. The values could be monitored in real time via WIFI connection, and it was decided to manually turn AMICA off to prevent damage. AMICA was turned back on only just before
take-off so that the ICOS Enclosure temperature and consequently the spectral signal only fully stabilized well into the





flight. During flight, the air stream around the instrument proved sufficient to prevent Power Box overheating. Post campaign tests revealed the major part of the heat load inside the power box to come from 2 x 60 W heat dissipation by two boards controlling the TEC assemblies regulating enclosure temperature. The problem of overheating on the apron was remedied by the selection of different TEC controller boards (D7 a/b in Table 2) with a maximum heat dissipation of ~10 W at

maximum load each.

## 5.2 M55 Geophysica, Kathmandu, 2017

A second deployment on the M55 Geophysica was also part of the StratoClim project and is described by Stroh et al. (2020). Eight research flights took place from Kathmandu, Nepal, between 27 July and 10 August 2017 to analyse the chemical composition as well as transport and mixing processes within the Asian summer monsoon (ASM) anticyclone. AMICA data

for all flights are available (see *data availability* below) and a scientific analysis of vertical transport within the ASM anticyclone that uses AMICA CO data has recently been published (von Hobe et al., 2020).

Some of the issues described in Section 5.1 were still not fully resolved. While the use of the high flow filters resulted in the nominal characteristics of the flow system described in Section 2.4 with pressure drops as shown in Figure 4, the reduced effective pressure at the inlet on the aircraft surface was unavoidable. This means that instrument precision worsens with

absorption signals as pressure drops above 16.5 km flight altitude. This is particularly relevant for the measurement of OCS that absorbs only weakly at atmospheric concentrations. Electrical noise was reduced compared to the 2016 campaign by rewiring a few particularly critical components (in terms of power draw variability, e.g. valves), but the comprehensive power and grounding concept described in Section 2.5 was only implemented after this campaign. Another issue encountered during the Kathmandu deployment was related to laser temperature control, which at the time was done by different control-

lers that received their set point from the PC via an analog 0 – 10 V signal. Both fluctuations and drifts were encountered that affected the position of the spectral window and introduced additional uncertainties in the assignment of the wavenumber scale to each measured spectrum, making the spectral analysis difficult.

## 5.3 HALO, SouthTRAC, 2019

The first AMICA deployment on the German HALO aircraft took place during the SouthTRAC mission in 2019 (Rapp et al.,

2020, also see https://www.pa.op.dlr.de/southtrac/). The payload was integrated in Oberpfaffenhofen, Germany, in July/August. The main campaign base was in Rio Grande, Argentina, where local science flights were carried out in two phases in September and November. Transfer flights with stops in Sal on the Cape Verde Islands and Buenos Aires, Argentina, were carried out in early September, early October and early November.

AMICA was situated in a rack near the front of the aircraft cabin. For SouthTRAC, it was flown with the fully implemented

power and grounding concept and a revised design of the ICOS control and data acquisition as described in Sections 2 and 3. Cavity pressure regulation worked nominally up to the HALO ceiling altitude of about 15 km. Except for two short test

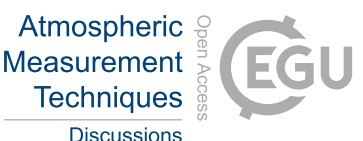

flights in Oberpfaffenhofen in August, where the recorded spectra were not correctly stored to disk due to a software problem, AMICA operated and recorded data during all flights. Towards the end of the long transfer flights at the beginning of the campaign, some data gaps are present that were caused by incidental shut offs of the LTC-1141 boards C6a and C6b to

prevent overheating. A close examination of this issue at the campaign base in Rio Grande revealed that the NTC sensors of the two large TEC boards (Figure 2) were incorrectly connected causing them to work against each other and not sufficiently cool the ICOS enclosure. By swapping the NTCs into their correct positions (as shown in Figure 2), the problem was solved.

Figure 13 shows observed trace gas mixing ratios and selected housekeeping data for the SouthTRAC flight on 12 Nov 2019 when AMICA operated nominally over the entire flight. The figure nicely illustrates the excellent stability of cavity pres-

sures (middle panel) and enclosure as well as QCL temperatures (bottom panel). Kloss et al. (2021) have used CO observations from this flight to identify a plume originating from bush fires in Australia.

## 6 Conclusions

With AMICA, an automated airborne OA-ICOS analyser has been developed that fully complies with all mechanical and electronic airworthiness requirements and that has successfully been deployed in three measurement campaigns on two very

different aircraft. A number of distinct or even unique design features are particularly innovative:

- A single instrument housing contains two discrete OA-ICOS cavities, allowing for the simultaneous measurement of a wider range of different species in two different wavenumber regions. The two ICOS systems share a common power supply system and control PC.

- By exchanging only cavity mirrors, laser source and detector, the target wavenumber region of each OA-ICOS

setup can be switched, and the suite of species measured by AMICA can to some extent be tailored to the scientific questions of a particular campaign. Where a particular campaign/aircraft protocol provides for temporary dismounting and modifications of instruments, the exchange of an OA-ICOS cavity in AMICA can even be carried out during a campaign within approximately one working day.

- The flow system with two parallel valves with different orifices provides for the precise regulation of cavity pres-

sures over the wide range of ambient pressure encountered between ground and maximum flight altitude.

- The containment of the sensitive OA-ICOS hardware in a pressure sealed and thermally controlled enclosure allows for deployment outside the aircraft cabin, i.e. in instrument bays exposed to the low pressure and low temperature conditions at flight altitudes up to at least 20 km. To seal the large rectangular, an adhesive was applied to the surfaces where individual plates were bolted together.

Further deployments of AMICA in future airborne campaigns are planned. Currently, work is being done to further improve the signal-to-noise ratio, e.g. by filtering the zero offset signal from the PC to the preamplifier. It is also planned to install mirrors with higher reflectivity and possibly an ICL with higher power output for the 3331 cm$^{-1}$ channel to render it suitable

for measuring $N_2O$, HCN and $C_2H_2$ at atmospheric concentrations. It is also envisaged to eventually add further channels at different wavenumber regions to AMICA in order to target additional trace gases, depending on the science questions of future missions.

*Code availability.* The PYTHON software code of the measurement programme running on the AMICA embedded PC will be made available prior to final publication of this paper.

*Data availability.* AMICA data from the two StratoClim Geophysica campaigns in 2016 and 2017 will be accessible via the HALO database at https://halo-db.pa.op.dlr.de/mission/101. Data from the 2019 SouthTRAC campaign will be accessible via the HALO database at https://halo-db.pa.op.dlr.de/mission/116.

*Author contributions.* C. Kloss and M. von Hobe coordinated the AMICA instrument development, carried out laboratory tests, operated AMICA during airborne campaigns and prepared the manuscript with contributions from all co-authors. V. Tan, J. B. Leen, G. L. Madsen, A. Gardner, X. Du, T. Kulessa, J. Schillings, H. Schneider and S. Schrade made major contributions to the AMICA conceptual design, development and construction. C. Qiu carried out laboratory tests and calibrations with the latest instrument version.

*Competing interests.* The authors declare that they have no conflict of interest.

## Acknowledgements

We would like to thank Gennady Belaev and the MDB team operating the M55 Geophysica aircraft and Andreas Minikin, Andrea Haushold and the DLR flight operations team responsible for the HALO aircraft for the successful deployments and their support with instrument integrations and operations, as well as local airport and ATC staff in Kalamata, Kathmandu, Oberpfaffenhofen and Rio Grande for their support. We also thank Rolf Maser, Dieter Schell and Jana Stahl from ENVISCOPE for their support of the AMICA integration on the HALO aircraft, and Nicole Spelten for helping with AMICA operations in Oberpfaffenhofen. We are grateful to Raffaele Sury from Meerstetter Engineering for his support and guidance when setting up the LTC-1141 for laser control and data acquisition. Marc von Hobe received support from the StratoClim project funded by the European Community's Seventh Framework Programme (FP7/2007-2013) under grant agreement n° 603557, the German Bundesministerium für Bildung und Forschung (BMBF) under the ROMIC-SPITFIRE project (BMBF-FKZ: 01LG1205). Corinna Kloss was supported by the graduate school HITEC of Forschungszentrum Jülich.

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





**Table 1:** Dimensions, weights and power draw of AMICA itemized for different parts and for the two different aircraft configurations.

| *Dimensions* | | |
|---|---|---|
| AMICA instrument: | 1050 x 435 x 355 mm | |
| With M55 mounting: | 1160 x 670 x 440 mm | |
| HALO rack: | 1420 x 650 x 550 mm | |
| ***Weights*** | | |
| OA-ICOS enclosure (without lid, adapters): | 91 | kg |
| Power box: | 16 | kg |
| Pump: | 4.1 | kg |
| **Total Instrument (without lid, adapters):** | **111** | **kg** |
| Enclosure lid for HALO: | 3.3 | kg |
| HALO rack: | 17.0 | kg |
| HALO rack mounting adapters: | 9.1 | kg |
| HALO rack power distribution box: | 3.7 | kg |
| **Total HALO configuration:** | **144** | **kg** |
| Enclosure lid for M55: | 8.7 | kg |
| M55 mounting hardware (springs and plates): | 12.6 | kg |
| **Total M55 configuration:** | **132** | **kg** |
| Handles and shackles for lifting: | 3.7 | kg |
| **Power Draw: typical/max** | | |
| PC + OA-ICOS system: | 350/400 | VA |
| Enclosure TEC assemblies: | 100/400 | VA |
| Pump: | 50/150 | VA |
| **Total instrument:** | **500/950** | **VA** |
| Inlet heating: | 230 | VA |




**Table 2** List of connectors and functional components in AMICA with information on specifications and purpose.

| # | Name | Make + Model | purpose/description/power |
|---|------|--------------|---------------------------|
| | | *Power Box connectors* | |
| J1 | main AC power | Souriau 8D0-17W06PN | Connects external AC input |
| J2 | Spare | Souriau 8D0-13W08SN | *currently no purpose* |
| J3 | inlet heating | Souriau 8D0-13W08SN | power line + PT100 signal |
| J4 | cockpit feedback | Souriau 8D0-15W19PN | send status signal (only M55) |
| J5 + J6 | external fan power | Lemo | |
| S1 | PB - ENC power (→P1) | MIL-STD MS3470W22-41SW | internal connections transfer power and transmit signals from the Power Box to the Enclosure |
| S2 | PB - ENC signals (→P2) | MIL-STD MS3476W22-55SW | |
| S3 | pump power | Lemo | |
| | | *Enclosure connectors* | |
| P1 | PB - EMC power (→ S1) | Glennair 230-016FT22-41PW | internal connections transfer power and transmit signals from the Power Box to the Enclosure |
| P2 | PB - ENC signals (→ S2) | Glennair 230-016FT22-55PW | |
| USB1/2 | C1 USB ports | USB HERMETIC3529 | Can connect USB devices (memory, keyboard, mouse, etc.) |
| ETH1 | C1 Ethernet port | RJ45 HERMETIC3273 | Ethernet connection (e.g. external PC, aircraft network) |
| ETH2 | spare Ethernet port | | used for VGA extender |
| WIFIA | C1 Wi-Fi Antenna SMA | PE9184 1525 | allow Wi-Fi communication (needed for M55 operation) |
| | | *Power Box components* | |
| F1 + F2 | EMI Filter (AC) | Schaffner FN2090-20-08 | minimize EMI interaction between AMICA and aircraft power system |
| F5 + F6 | EMI Filter (DC) | CUI INC VFM-15C | filter out electrical noise on DC currents |
| F9 | MicroRAM | VICOR | actively filter 24 V DC power supplied to laser drivers C6a and C6b |
| B1-5 | Push button thermal breakers | eta-483, 2.5/1/2/6/4 A | AC power side breakers |
| V1 | AC/DC converters | VICOR, VP-G3001410E with HUB1800-S and FZJ designed fuse/current monitor board | **IN: 90 - 270 VAC   OUT: 24 V DC (1 channel)** power pump D8 and external fans |
| V2 | | VICOR, VP-C2916325E with HUB3300-S and FZJ designed fuse/current monitor board | **IN: 90 - 270 VAC   OUT: 24 V DC (2 channels)** power TEC controllers D7 |
| V3 | | | **IN: 90 - 270 VAC   OUT: 24 V DC (2 channels)** power enclosure components (Section 2.5 for details) |
| D1 | DC/DC converter | CUI INC VHK100W-Q24-S5 | **IN: 24 VDC from V3 CH 1  OUT: 5 VDC** supply PC (C1) and SSD (C2) via distribution board C13a |
| D2 | DC/DC conversion board | FZJ, with 2 TRACOPOWER TDR 2-2422WI | **IN: 24 VDC from V3 CH 1  OUT: 2 x ± 12 VDC** separate DC/DC modules provide galvanically separated power to PreAmps (C9) |
| D3 | DC/DC converter | CUI INC VHK200W-Q24-S12-DIN | **IN: 24 VDC from V3 CH 2  OUT: 12 VDC** supplies 12 V power to C1, C2, C3, C4, C10, C11, C12 |
| D4 | Temperature board | FZJ, with 1 TRACOPOWER TDR 2-2422WI | **IN: 24 VDC from V3 CH 1  OUT: 5 VDC / ±12 VDC** operate sensors monitoring pump and VIPAC temperatures (5 V); DC/DC converter supplies ± 12 V to cavity temp. sensors C14a/b |





| D5 | Fuse board | FZJ | fuses protect individual power lines from excess current |
|---|---|---|---|
| D6 | Data logger | LabJack T7 OEM + connector board | **IN: 5 VDC from USB**<br>monitors housekeeping parameters in the Power Box (temperatures, voltages, currents) and sends data to C1 via USB. |
| D7 a/b | TEC controller | Meerstetter TEC-1189-SV | **IN: 24 VDC from V2 CH1**<br><br>Control and supply current to the two enclosure TEC assemblies |
| D8 | Diaphragm pump | Vaccuubrand MD1 Vario | **IN: 24 VDC from V1**<br>draws sample air from the inlet through both cavities in series (see Section 2.4) |
| D9 | Solid State Relay | CRYDOM 4D2425 | **IN:  main AC**         Relay works together with controller D10 |
| D10 | Heater controller | OMEGA CN32Pt-440 | **IN:  main AC**         controls inlet temperature via heater |
| PBFAN1/2 | external TEC bank fan | ebm Papst VarioPro 4114 NHU | **IN: 24 V from V1** |
| *Enclosure Components* | | | |
| C1 | embedded PC/104 stack | ADVANTECH PCM 3363 + Redwave S310 + RTD LAN18222HR + RTC WLAN18202ER | **IN: 5 VDC from D1**<br>Embedded PC handles<br>  - data acquisition, storage and spectral analysis<br>  - communications |
| C2 | SSD (64 GB) | Transcend TS64GSSD25S-M | **IN: 5 VDC from D1**<br>hard disk, separated from PC 104 stack for easier service access |
| C3a/b | pressure controller | Redwave | **IN: 12 VDC from C13a/b**<br>Proportional controllers (see Section 2.4 for details) |
| C4a/b | pressure gauge | Honeywell | **IN: 12 VDC from C13a/b**<br>Measure cavity pressures. |
| C5a | valve | Parker EPCA55SSVCAA 0.7 mm | Proportional solenoid valve operated by C3a |
| C5b | valve | ASCO Posiflow SCB202A013V12VDC 3.2 mm | Proportional solenoid valve is operated by C3b |
| C6a/b | laser controller board | Meerstetter LTC-1141 | **IN: 24 VDC from V3 CH1**<br>LTC boards control laser TEC + current, 200Mbit ADC channels read signals from C9a/b (see Section 2.2) |
| C7a/b | laser diode + mount | Opts. according to Table 3 | laser emitting light into cavity, controlled by C6a/b |
| C8a/b | detector + mount | Opts. according to Table 3 | converts light exiting the cavity into current signal passed on to C9a/b |
| C9a/b | preamplifier | Femto, HCA-S2 or DLP-CA-200 | **IN: ±12 VDC from D2**<br>convert C8 currents to voltage + amplify; zero adjust via 0 - 10 V from C1 |
| C10a/b | 2 channel TEC controller | Meerstetter TEC-1122 | **IN: 12 VDC from C13a/b**<br>regulate laser mount (C7) and detector mount (C8) temperatures |
| C12 | 6-way PZT driver | LGR designed | **IN: 12 VDC from C13a** |
| C13a | power distr. board 12V + 5V | FZJ | distributes 5V from D1 to C1, C2 and 12 V from D3 to C3a, C4a, C10a, C12; onboard P and T sensors monitor enclosure pressure and temperature |
| C13b | power distr. board 12V + 24V | FZJ | distributes 12 V from D3 to C3b, C4b, C10b and 24V from V3 CH2 to fans |
| C14a/b | Cavity T sensor | LGR designed | Measure cavity temperatures. |
| Fan1/2 | Internal fans TEC banks | ebm Papst VarioPro 4114 NHU | **IN: 24 VDC from C13b**<br>heat distribution for enclosure temperature homogenization |
| Fan3/4 | fans enclosure wall | ebm Papst 3414 NHU | |
| Fan5/6 | fans laser housing | ebm Papst 3414 NHU | **IN: 24 VDC from C13b**<br>heat transport from thermo-regulated components |
| Fan7 | fan detector C8a | NMB, 2406KL-05W-B50-L00 | |
| Fan8 | fan detector C8b | Multicomp, MC36321 | |





**Table 3** Currently implemented ICOS cavity configurations. Each measurement setup consisting of laser and driver board, dielectric mirror and detector can be used in either one of the two cavity positions in AMICA.

| | ν range in cm$^{-1}$ | gases @ line position in cm$^{-1}$ | Laser type, model, max. output | Detector type and model | Mirror $R$ |
|---|---|---|---|---|---|
| **I** | 2050.23 – 2051.47 | OCS @ 2050.4<br>CO$_2$ @ 2050.57<br>CO @ 2050.85<br>H$_2$O @ 2050.64 | QCL<br>Hamamatsu<br>55 mW | HgCdTe photodiode<br>Teledyne Judson<br>J19TE3:5-66C-R01M<br>(1 mm apt., -65°C, 4.5 μm) | ~ 0.9998 |
| **II** | 1033.21 – 1034.36 | O$_3$ @ 1033.68<br>NH$_3$ @ 1033.32 | QCL<br>Hamamatsu LC0026<br>40 mW | Photovoltaic Mult. Junct.<br>VIGO System<br>PVMI-4TE | ~ 0.9995 |
| **III** | 3330.8 – 3332.0 | HCN @ 3331.59<br>C$_2$H$_2$ @ 3331.34<br>N$_2$O @ 3331.65 | ICL<br>Nanoplus<br>8 mW | HgCdTe photodiode<br>Teledyne Judson<br>J19TE4:3-5CN-R01M<br>(1 mm apt., -80°C, 4.5 μm) | ~ 0.995 |

**Table 4** List of housekeeping parameters routinely monitored during AMICA operation. Parameters marked by bold face are sent to the HALO Web User Interface PLANET (ATMOSPHERE, Wessling, Germany) for real time online monitoring.

| Type | sensor description/location | acquisition |
|---|---|---|
| *Power Box* | | |
| voltages and currents | • V1 add on board (two channels)<br>• V2 add on board (two channels)<br>• V3 add on board | analog signals for all parameters are digitized by the LabJack T7 data logger (D6) and transmitted to the embedded PC (C1) via USB |
| **Temperatures** | • **V1**<br>• **V2**<br>• **V3**<br>• **pump** | |
| *Enclosure* | | |
| enclosure pressure and **temperature** | C13a | analog signals are acquired by a RedWave S310 data acquisition card in the PC/104 stack |
| **cavity pressures** | **C4a and C4b** | |
| cavity temperatures | C14a and C14b | |
| LTC and laser parameters (separate for LTC boards C6a and C6b) | • **QCL/ICL temperature**<br>• laser heat sink temperature<br>• laser TEC current<br>• LTC board + CPU temperatures<br>• LTC supply voltage | communicated from each C6 board to the PC via UDP data stream (one set of parameters per averaged spectrum) |





**Table 5** Compilation of bottled standards and procedures to prepare standards used for laboratory tests and calibrations of the AMICA setups described in this paper.

| Gases | AMICA setup (as in Table 3) | Standard composition or preparation procedure |
|---|---|---|
| $CO_2$, CO | I | Bottled standard (50 dm³, 200 bar), Air Products |
| | | Composition: 5000 ppm (± 0.5 % rel.) $CO_2$, 5 ppm CO (± 1 % rel.), 5 ppm (± 1 % rel.) $N_2O$, 25 ppm (± 0.5 % rel.) $CH_4$ in $N_2$ |
| | | Dilution:  $N_2$ 6.0 (99.9999 % purity) with MFCs (Natec MC-10SLPM-D and MC-50SLPM-D, accuracy < 0.4 % absolute) |
| OCS | I | 1. Permeation device (emitting 26.1 ± 0.1 ng/min, determined by regular weighing) held in Sulfinert treated flow chamber (at 25.000 ± 0.005 °C); Concentration is set via the flow rate, controlled with MFCs (Natec MC-10SLPM-D and MC-50SLPM-D, accuracy < 0.4 % absolute). See von Hobe et al. (2021) for more details on the calibration system<br>2. NOAA standard: 61 atm natural air in electropolished stainless steel cylinder, containing 449.8 ± 1.4 ppt OCS determined by GCMS |
| $H_2O$ | I | relative humidity sensor (MSR 165) |
| $O_3$ | II | $O_3$ generator with integrated UV-Photometer as reference (Proffitt and McLaughlin, 1983) |






**Figure 1** Technical drawing of AMICA showing bulk parts. Additional drawings including M55 mounts and the HALO rack are given in the supplementary Figures S1 and S2 respectively.

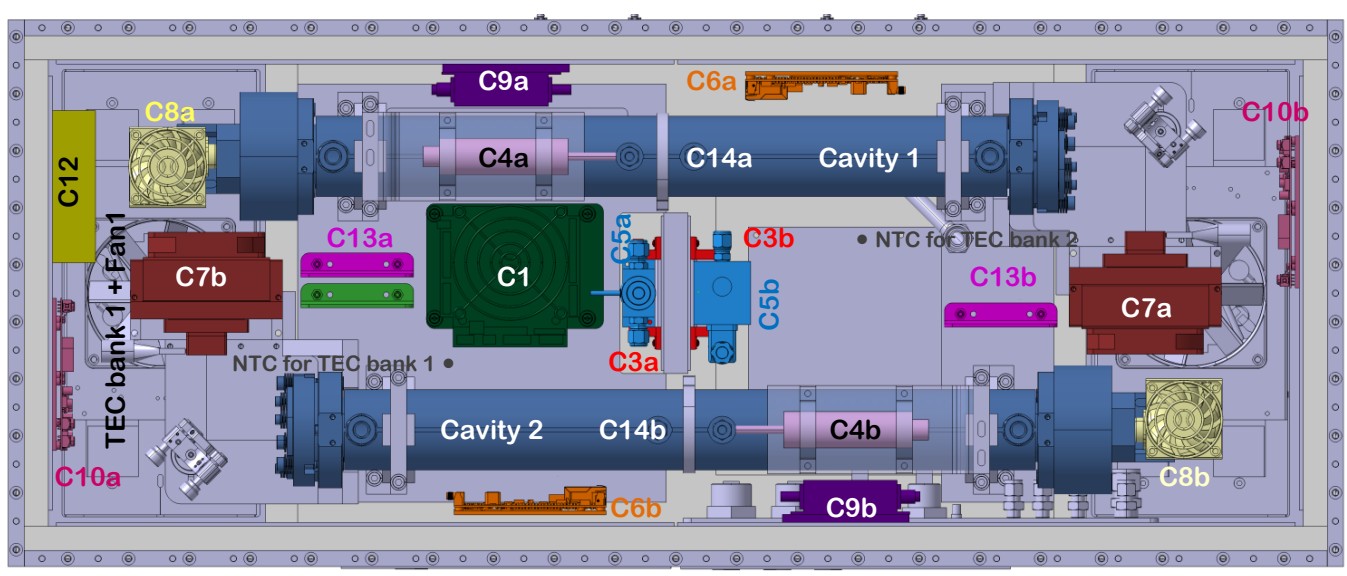


**Figure 2** ICOS arrangement inside the AMICA enclosure. Labelled components are described in Table 2.





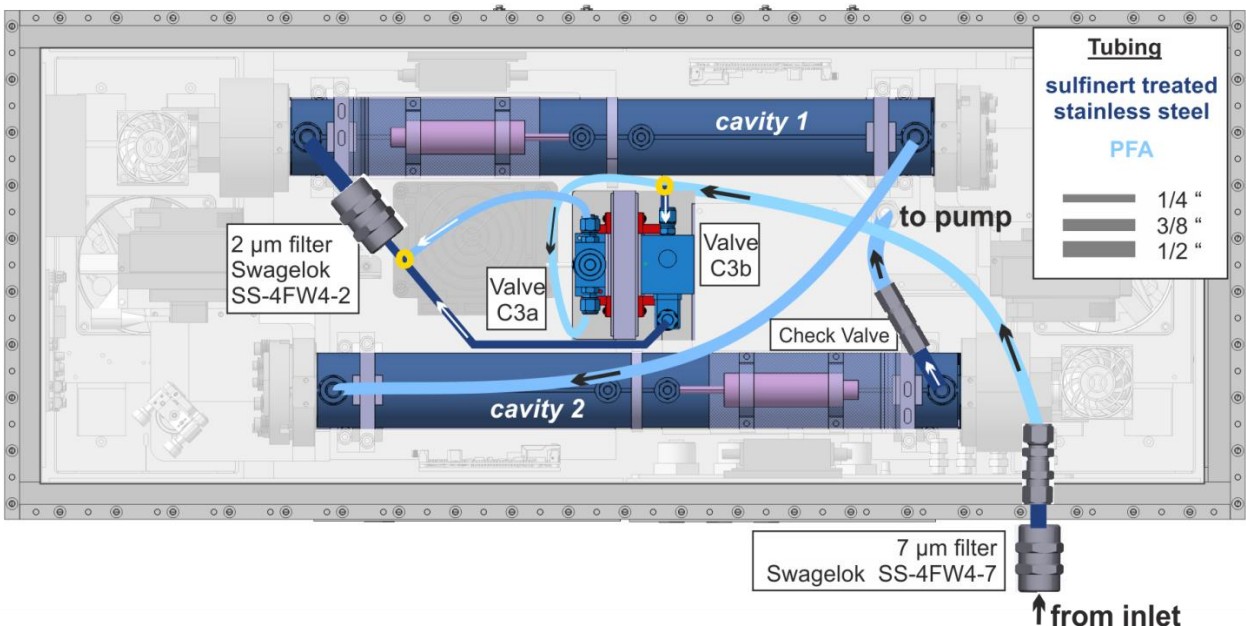

**Figure 3** Sample gas flow system inside AMICA. The T-junctions where the gas flow branches off to/from the two valves C3a and C3b
are marked by yellow circles.





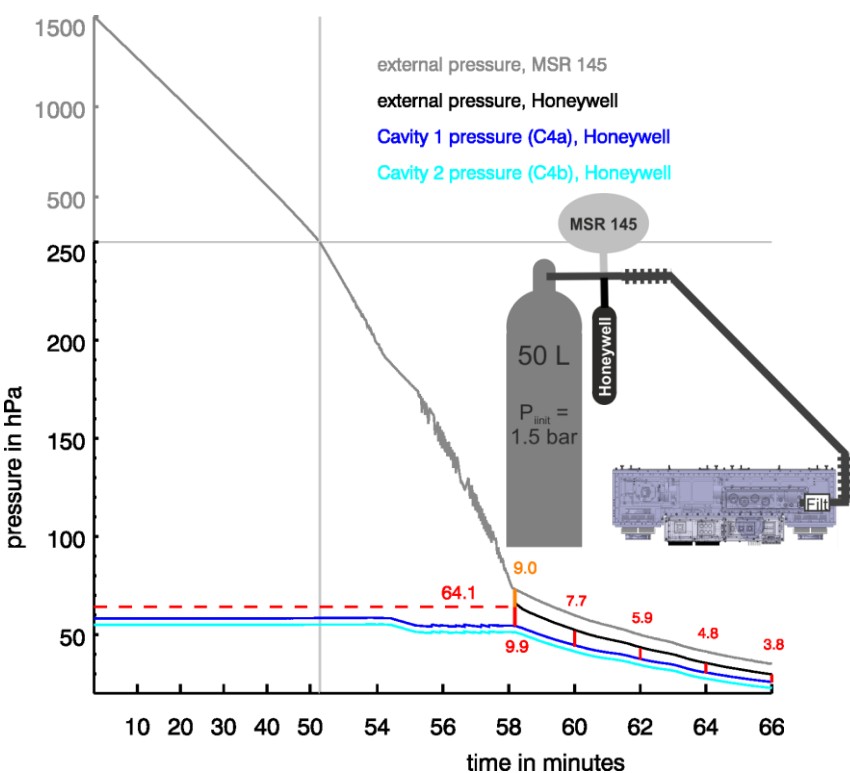

**Figure 4** Experimental determination of AMICA cavity pressures (dark blue: cavity 1; light blue: cavity 2) to simulated ambient pressure at a 50 L gas bottle measured by the pressure gauge of an MSR 145 data logger (grey, absolute range 0 – 2000 hPa, certified accuracy of ± 2.5 hPa between 750 and 1100 hPa) and a Honeywell gauge identical to the gauges C4 inside AMICA (black, range 0 – 69 hPa). At 64.1 hPa, the offest between the two gauges is 9.0 hPA (shown in orange); at pressures within the range of the Honeywell gauge, we deem this sensor more accurate than the MSR 145 and more comparable to the C4 gauges inside AMICA. For the regime where cavity pressure cannot be regulated, differences between simulated ambient pressure at the 50 L bottle and cavity 1 pressure (measured by gauge C4a) are shown in red. Note that the scaling of the x and y axes changes at 52 min and 250 hPa respectively as shown by the light grey lines.





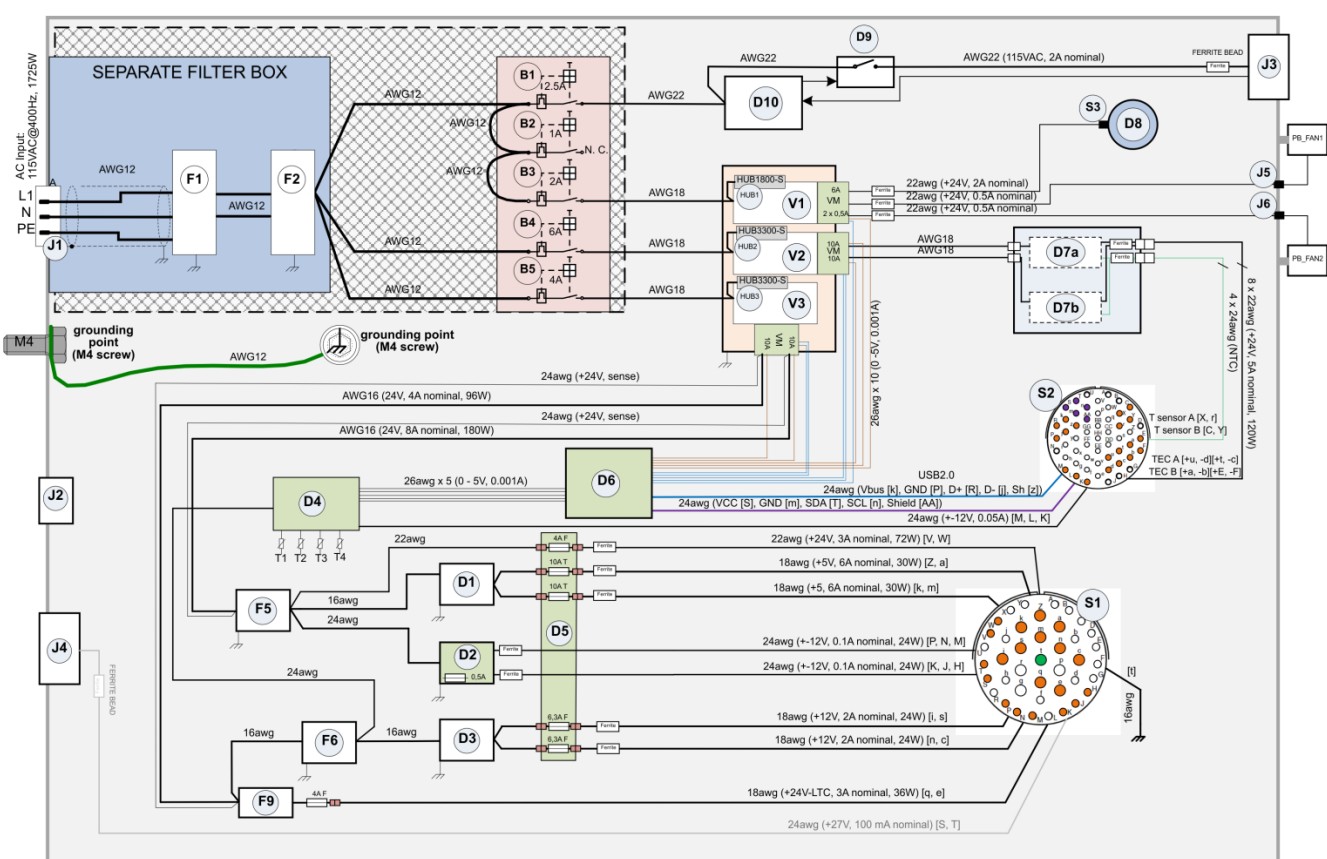

**Figure 5** Simplified block diagram of the AMICA power box. Component and connector positions and sizes are not drawn to scale, identifyers correspond to those used in Table 2. Wire gauge is shown for all lines, voltage and nominal current is also given for power lines.



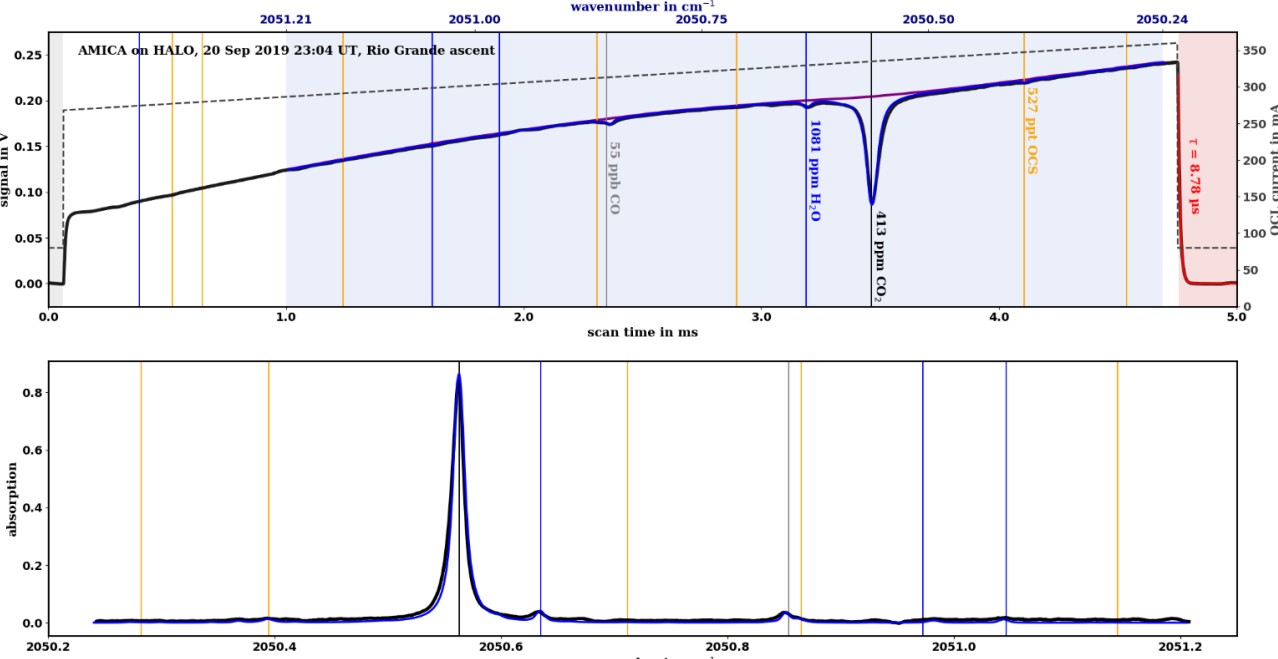

**Figure 6** Top: QCL current (dashed line, right axis; the 268 – 360 mA ramp is slightly curved to arrive at the linear wavenumber scale indicated on the top x-axis) and detector voltage signal past the preamplifier (solid black line, left axis; ramps averaged over 30 seconds) for AMICA cavity 1 in the 2050 cm$^{-1}$ setup. Three periods are marked by the colored areas: (i) grey: only detector dark signal (with zero offset) is observed; (ii) blue: period used for spectral analysis; (iii) red: period after QCL stops emitting, when the signal decays exponentially due to "ring down" of the light in the cavity. A ringdown fit is shown by the red line. For the spectral fitting region, a measured baseline spectrum with no absorbers present (purple) and a fitted spectrum (blue) are also shown. Bottom: measured and fitted spectra from top panel shown in absorption space.



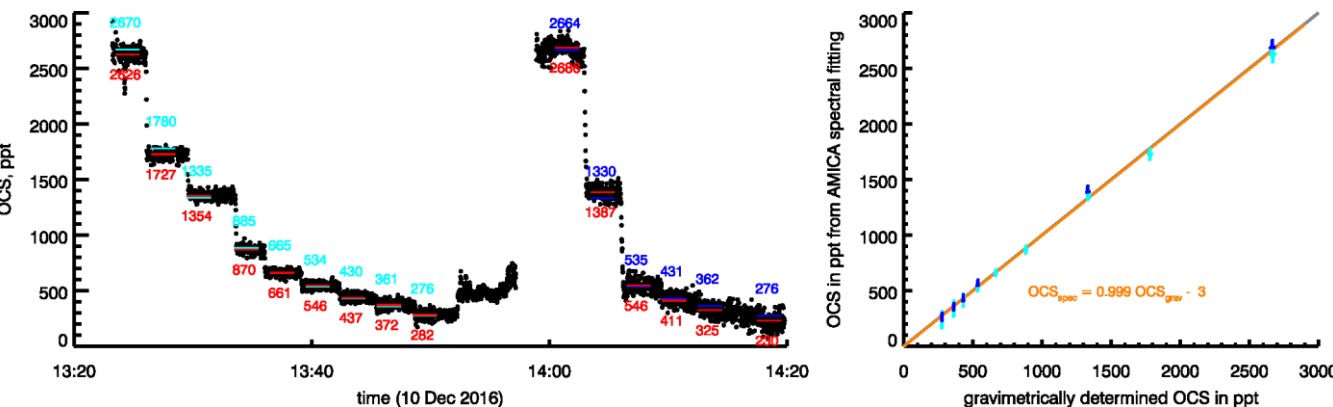

**Figure 7** Laboratory calibration for AMICA OCS. Left: OCS mixing ratios fitted to AMICA spectra in black, with red bars and numbers indicating averages for each mixing ratio level. Mixing ratios of the standards are also shown as bars and numbers for a series at $P_{cav}$ = 46 hPa (light blue) and one at $P_{cav}$ = 18 hPa (dark blue). Right: fitted AMICA OCS against the gravimetric standards with the same blue colors indicating $P_{cav}$ of each measurement. Error bars in x direction represent the uncertainty of the gravimetric standards (propagated from uncertainties in permeation rate and flow rates), error bars in y direction represent the statistic measurement uncertainty determined at each mixing ratio. The 1:1 line is shown in grey, and a linear fit to all data (independent of $P_{cav}$) in orange.

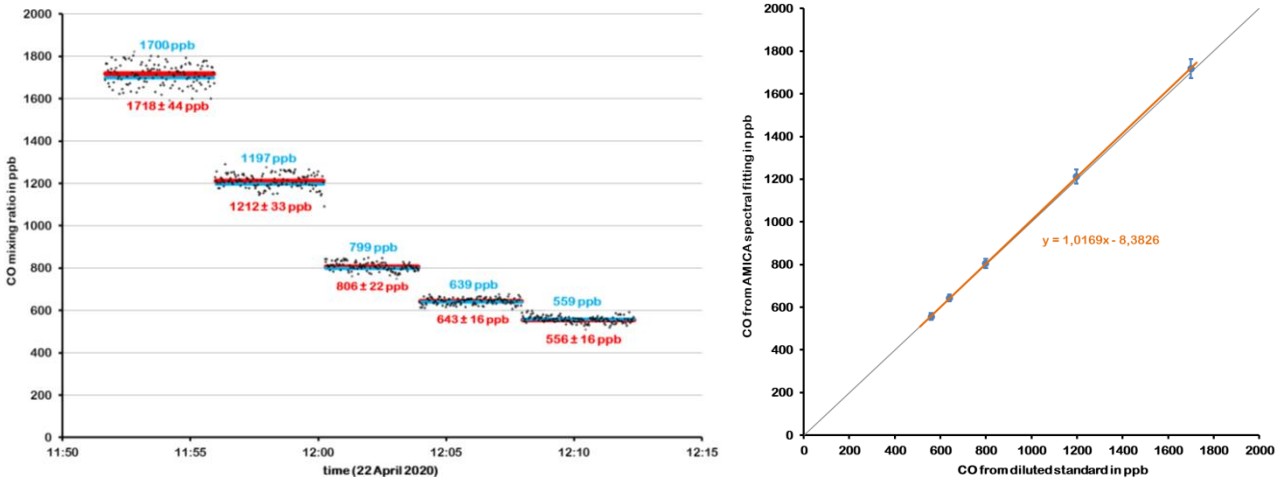

**Figure 8** The left panel shows the CO mixing ratios fitted to AMICA spectra (black: individual data points; red: averaged with standard deviations) and the known mixing ratios in the standards (light blue) against time of the experiment. The right panel shows AMICA CO vs. standard CO and a linear fit of these data.





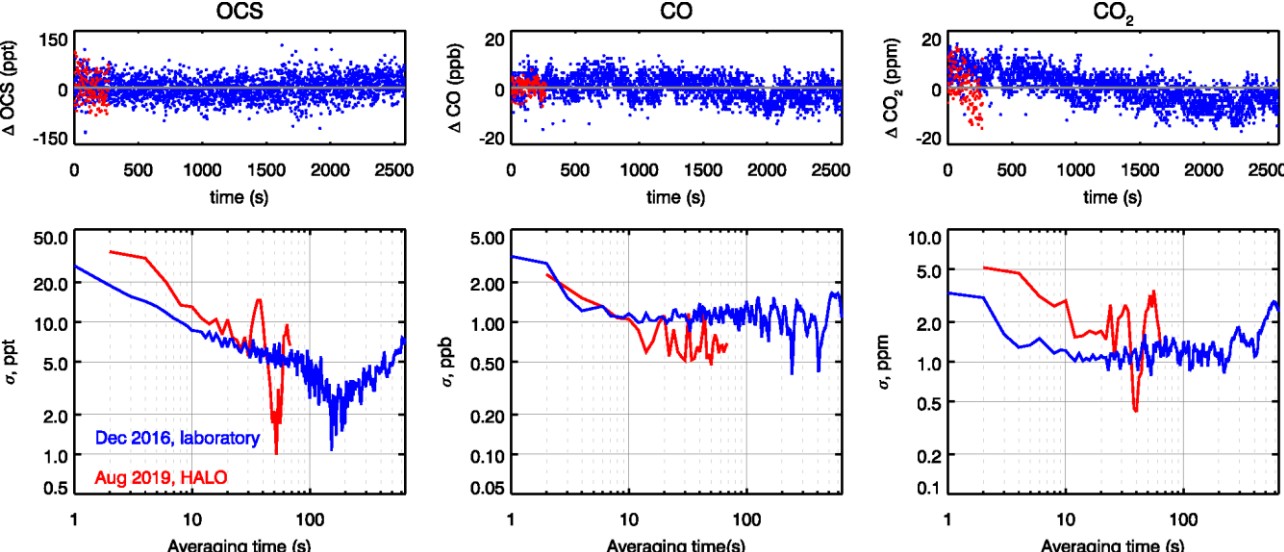

**Figure 9** Allen Deviation plots for OCS, CO and $CO_2$ based on measurement of the same standard by AMICA over an extended period of time (45 minutes with 1 second time resolution in the laboratory test in Dec 2016, blue line, and 5 minutes with 2 second time resolution inside the HALO aircaft at the start of the SouthTrac campoaign).



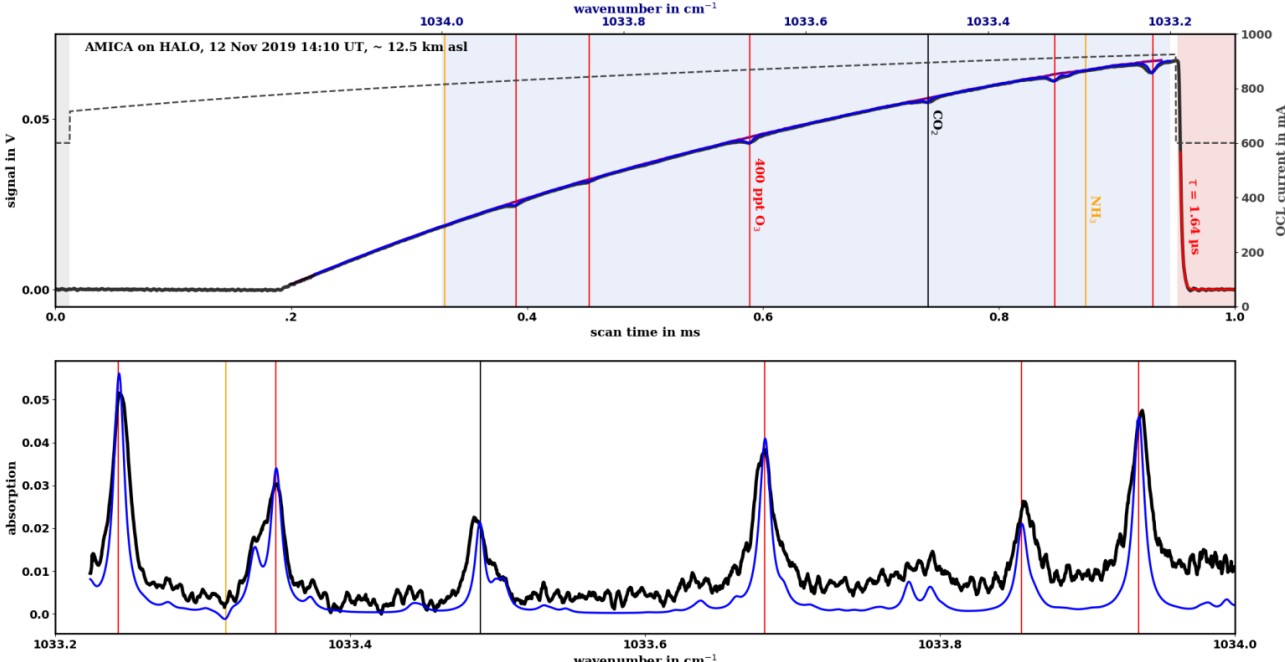

**Figure 10** Top: QCL current (dashed line, right axis; the 775 – 925 mA ramp is slightly curved to arrive at the linear wavenumber scale indicated on the top x-axis) and detector voltage signal past the preamplifier (solid black line, left axis; ramps averaged over 30 seconds) for AMICA cavity 2 in the 1034 cm$^{-1}$ setup. Three periods are marked by the colored areas: (i) grey: only detector dark signal (with zero offset) is observed; (ii) blue: period used for spectral analysis; (iii) red: period after QCL stops emitting, when the signal decays exponentially due to "ring down" of the light in the cavity. A ringdown fit is shown by the red line. For the spectral fitting region, a measured baseline spectrum with no absorbers present (purple) and a fitted spectrum (blue) are also shown. Bottom: measured and fitted spectra from top panel shown in absorption space.






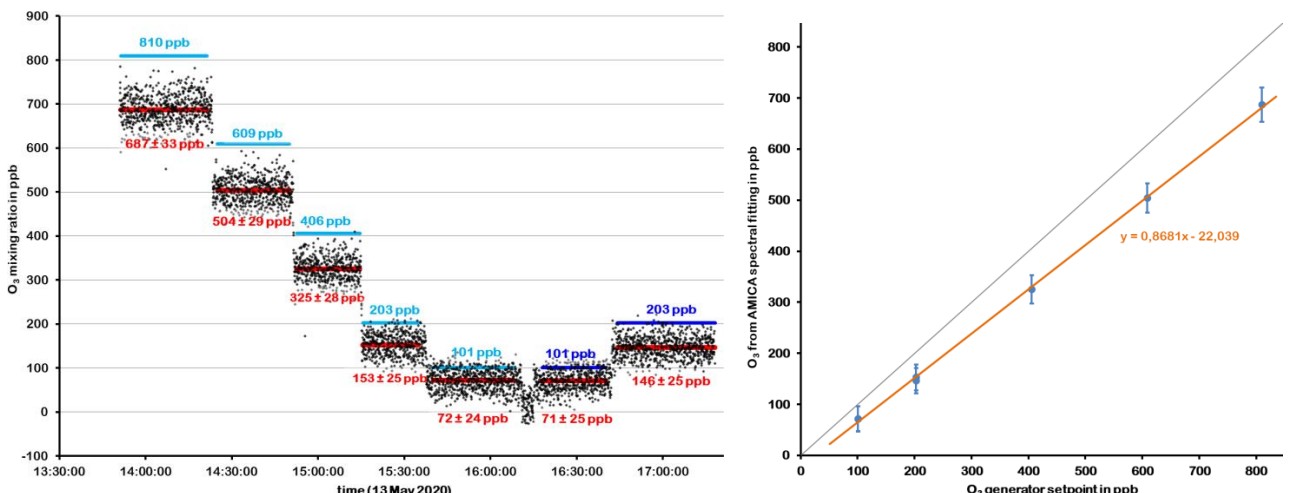

**Figure 11** The left panel shows the $O_3$ mixing ratios fitted to AMICA spectra (black: individual data points; red: averaged with standard deviations) and the known mixing ratios in the standards (light/dark blue: with/without stainless steel tube in transfer line, see text for details) against time of the experiment. The right panel shows AMICA $O_3$ vs. standard $O_3$ and a linear fit of these data.




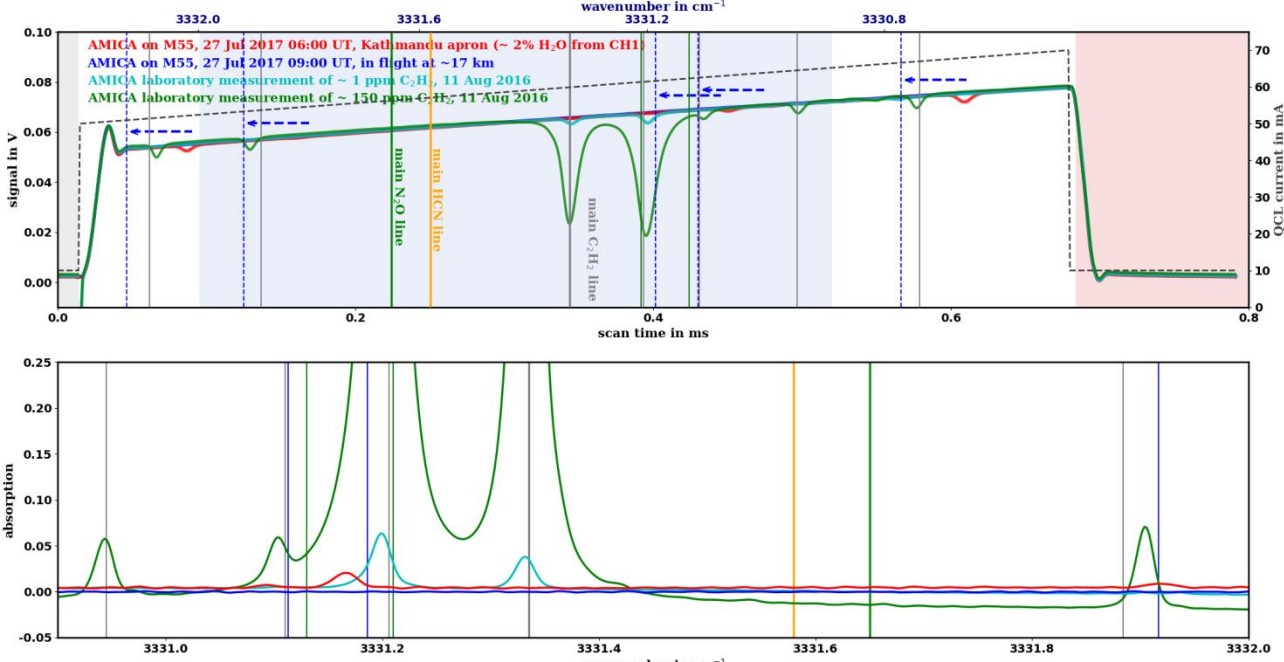

**Figure 12** Top: QCL current (dashed line, right axis) and detector voltage signal past the preamplifier (left axis; ramps averaged over 300 seconds) for AMICA cavity 2 in the 3331 cm⁻¹ setup for a laboratory experiment sampling $C_2H_2$ from a bottle and a flight from
Kathmandu (see legend for colors). The wavelength scale shown on the top axis was fitted to the laboratory data. The wavelength scale during the flights was shifted, so that the observed $H_2O$ peaks are relatively displaced (indicated by the dashed blue arrows). Three periods are marked by the colored areas: (i) grey: only detector dark signal (with zero offset) is observed; (ii) blue: period for which absorption spectrum in the lower panel is shown; (iii) red: period after QCL stops emitting, when the signal decays exponentially due to "ring down" of the light in the cavity (the ringdown was to fast to do a reasonable fit). Bottom: measured spectra from top panel shown in absorption
space (with the wavelength scale corrected).





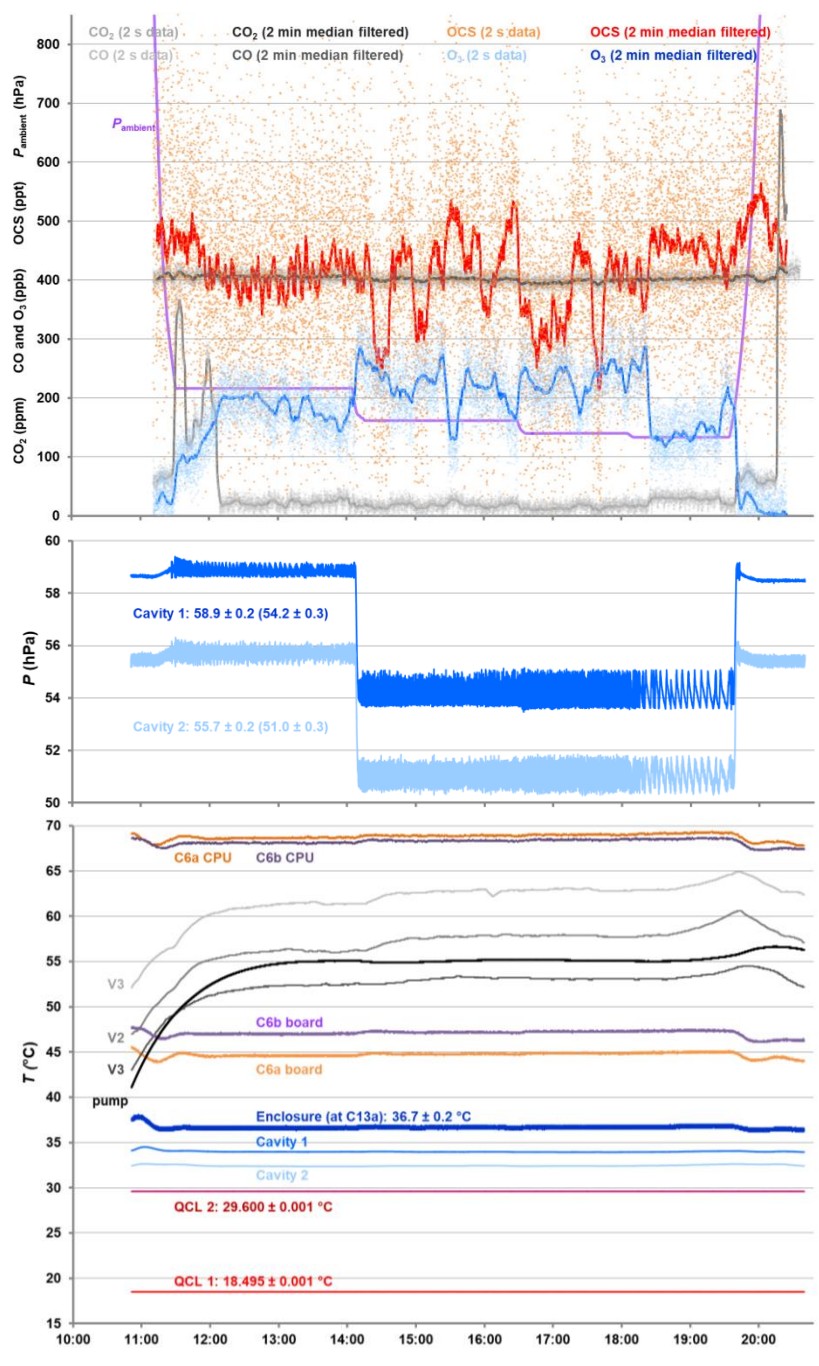


**Figure 13** AMICA trace gas observations (top), cavity pressures (middle) and selected temperatures (bottom) for the SouthTRAC flight on 12 Nov 2019 from Rio Grande, Argentina.