# Peer review of "Airborne Mid-Infrared Cavity enhanced Absorption spectrometer (AMICA)"

_Atmospheric Measurement Techniques, 2021_

## Author Comment (AC1)

**Response to 'Comment on amt-2021-28' by Frans Harren**

*We thank Frans Harren for his constructive review of our AMICA manuscript. He identified some obvious mistakes and shortcomings and made important suggestions how to improve the paper. Below, we copy Frans Harrens comments in black and give our response to each of them in blue font. Significant changes to the text in the revised manuscript are shown here in purple font.*

This is an interesting manuscript describing the technical set-up and calibration experiments of the AMICA instrument with demonstrating results for various aircraft campaigns. The technical part is described well, while the performance of the instrument and the molecular spectral interpretation could be better explained. As such I recommend substantial changes to the manuscript before accepting it for publication.

- In the abstract precisions are given for the various molecules, but within the text these precision data are not convincingly confirmed.

The discussion on accuracy and precision has been expanded both in the abstract and in the text, also in response to referee #2. The precisions given for OCS and CO in the abstract are now also given in the text together with a description on how they are derived from the Allen Plots in Figure 9:

"For the OCS and CO observations made so far, respective precision estimates of 30 ppt and 3 ppb are made based on the higher two second value from the two experiments shown in Figure 9. For $CO_2$, both curves exceed 1 ppm for all averaging times, which is another reason (besides the issues described above) for currently not releasing AMICA $CO_2$ data for scientific use."

- Line 77 the description of the ICOS set-up refers to Fig.2. This an informative figure, but fails to give clearly optical beams paths from laser to detector.

The paths of the optical beams are included in a revised Figure 2.

- Line 258, the trace gas inlet is rear facing, considering the pressure problems at high altitudes (line 560), it could be better discussed why the inlet is rear facing. Taking into account the position is of the inlet on the aircraft cabin, distance of inlet to the cabin, boundary layer around the plane at high altitudes. Why not using a Pitot-like tube, taking advantage of the dynamic pressure?

Rear facing inlets are common for gas phase measurements, because they provide an easy way to discriminate against the inflow of particles. While particle discrimination has also been realized with forward facing inlets that allow for passive sampling by making use of the ram pressure (e.g. Gao et al., AMT, 2012), the implementation of such inlets is more complex, especially when they are shared by multiple instruments.

We add the following information to Section 2.4 in the revised manuscript:

"Both inlet systems are rear facing to avoid the intake of liquid water, ice and large aerosol particles (McQuaid et al., 2013). They are briefly described and characterized here."

- A proper Figure (picture) of the inlet and transfer line will be helpful.

A link is added to the HALO webpage, where the TGI is described in much detail:

https://www.halo.dlr.de/instrumentation/inlets/inlets.html#TGI

The following figure of the AMICA inlet on M55 Geophysica is added to the supplementary material:

[Figure]

Line 130, next to the temperature control of the instruments, it is helpful to describe the heat transport resistance value of the foam, used on the ins of the enclosure walls, to keep the temperature of the AMICA within limits.

Relevant specs of the foam are added in Section 2.3 of the revised manuscript:

"density: 2.2 kg m$^{-3}$, thermal conductivity:  0.06 and 0.05 W m$^{-1}$ K$^{-1}$ at 24 and -5 °C respectively"

- Line 460 and below: I to have objections with the word "absorption band" when an absorption line is meant. Maybe except for ozone all measured absorption features are single molecular lines.

This is, of course, correct, and the terminology is corrected throughout the revised manuscript.

- Line 460: The ICOS description for measuring OCS, $CO_2$, CO and $H_2O$, is not convincingly describing the performance of the instrument. The mirror reflectivity is given in Table 3, what is the effective optical path length related to this? Taken the line strength and broadening effects into account and the noise of the detector, what are the expected detection sensitivities for these gases. How are these related to the real detection sensitivities (e.g. Fig. 7, 8) under lab conditions and flying conditions. What is the Noise Equivalent Absorptions Sensitivity of the instrument (see: Paul et al.)?

The effective optical path length $L_0 = L_{cav} / (1 - R)$ is added to Table 3 for each channel.

Laser, detector and mirrors for the 2050 cm$^{-1}$ channel are the same as in the enhanced performance model of the commercial OCS analyzer by Los Gatos that has nominal 1 Hz precisions of 16 ppt OCS, 0.7 ppb CO and 0.4 ppm $CO_2$ and a NEAS of about 1 x 10$^{-8}$ cm$^{-1}$ Hz$^{-1/2}$.

Currently, AMICA sensitivity is significantly reduced (by roughly a factor of 10) due to the electrical noise caused by the ground loop at the preamplifier as explained in the paper. Once that issue is resolved, we expect sensitivity in the same range as the commercial analyzer (this was verified by measuring the detector signal with an external oscilloscope).

Note that the electrical noise is the same under lab and flying conditions. Because of the effective EMI isolation of the instrument electronics from the surroundings, we also expect the detector noise to be the same. The one thing that does get noisier under flying conditions particularly at higher altitudes is the cavity pressure (cf. Figures 4 and 13), but the effect on overall sensitivity should be small.

- The same clarifications should be made for the other two wavelength regions described in 4.2 and 4.3.

For the 1034 and 3331 $cm^{-1}$ channels, a proper determination of sensitivity and precision has not yet been carried out because other issues will have to be resolved first as described in Sections 4.2 and 4.3.

For the 1034 $cm^{-1}$ channel, however, a first estimate of precision for $O_3$ can be made based on the experiments shown in Figure 11. This is now stated in the text:

"Based on the measurements shown in Figure 11, 0.5 Hz precision for $O_3$ is currently in the range of 20 to 40 ppb."

For $NH_3$, we reference once more the paper by Leen et al. (2013):

"Ultimately, it should be possible to measure $NH_3$ with similar sensitivity as described by Leen et al. (2013)."

For the 3331 $cm^{-1}$ channel, the low mirror reflectivity and effective path length do not allow measurements at atmospheric concentrations, so there is little point in giving precision data. Also following the suggestion of reviewer # 2, we stress in the revised manuscript that this is currently more a conceptual channel.

- Line 475: A discussion is made about the compromise of Leff, for strong $CO_2$ absorption and weak OCS absorption, and that the broadened $CO_2$ peak reduces sensitivity. It is unclear what the outcome of this discussion was and why a broadened $CO_2$ peak should reduce sensitivity?

This is already illustrated and described in some detail in the supplementary material (Figure S5).The sensitivity reduction a direct result of $L_{eff}$ becoming a quasi-inverse function of $A$ at high $A$ and, so that the scaling of $A$ with $c$ in Eq. (2) becomes significantly less than linear. The additional broadening results from the sensitivity reduction being stronger near the centre of the absorption peak and less at the flanks. Unlike with Doppler and pressure broadening, this "cavity broadening" behaves in a way that peak area does not scale linearly with concentration anymore.

To make this clearer in the text, we change the end of the paragraph in Section 4.1 to:

"The consequence is a smaller change in absorption for a given change in concentration and thus a reduced sensitivity. Because $L_{eff}$ varies with absorption over the broadened peak ($L_{eff}$ is smaller at the peak centre where A is larger), the effect also introduces additional broadening that complicates the analysis. More details and an illustration on the sensitivity reduction for $CO_2$ are given in Figure S5 in the supplementary material."

- Line 480, a fit is made; is this a Voigt fit?

Indeed, a Voigt fit is made. This information was "hidden" in line 454, where it was stated that Voigt profiles are calculated from the HITRAN parameters. In the revised manuscript, we also add the word "Voigt" before fit in line 480.

- Line 536, a low mirror reflectivity is chosen for the setup, although HCN, $C_2H_2$ and $N_2O$ concentrations are too low to be observed. What argument is made to install this low mirror reflectivity?

This highly experimental setup was implemented in a very short time frame for the StratoClim field campaigns. Because time was lacking to acquire and test an optimized HR mirror for this channel, a 3 µm mirror that had been sitting on the shelf was used, which unfortunately turned out to have a far too low reflectivity (clearly lower than we thought it would be).

When we chose to install this setup for the campaign, it was clear that we would not make measurements with this channel as we would make them with the operational 2050 cm$^{-1}$ channel. However, we had hoped for a somewhat higher reflectivity and thus path length that would allow us to see a signal for elevated monsoon HCN mixing ratios when averaging spectra over longer time periods. As described in the paper, we ended up only observing laser ramps with absorption due to high water vapour at the ground, and can thus only demonstrate some sort of spectroscopy being done but anything close to a measurement of our target gases HCN, $C_2H_2$ and $N_2O$.

Note that it is likely that with sufficiently reflective mirrors, the laser power of 8 mW will be too low to couple enough light into the cavity and onto the detector. Lasers with up to 25 mW at 3 µm have now become available (see for example https://nanoplus.com/en/icl/) and we still hope to eventually acquire a set of mirrors with sufficient reflectivity (up to at least 0.9995 is available) and a new ICL with sufficient output power to turn the setup into an operational channel.

- Figure 10 lower panel, the fit deviates quite from the experimental observation, giving a systematic offset in the mixing ratio fits (Fig 11). Observing the spectrum, it seems that there is broadband background change in absorption over the wavelength range. Such a change is not taken into account in the fit? This would even make the fitted ozone concentrations lower than the standard.

The following discussion on the obviously poor fit to the ozone spectrum is added to the first paragraph of Section 4.2:

"The spectral fit does not closely reproduce the observed spectrum. First, significant absorption up to 0.01 between peaks points to either a bias in the used baseline or trace gas absorption by lines or bands that are not included in the HITRAN data base. Second, the observed $O_3$ absorption peaks are broader than the fitted peaks. Possible explanations include inaccurate HITRAN parameters for the $O_3$ lines or cavity response broadening. Both, baseline offset and the broader peaks, will be further investigated in future laboratory experiments."

- Reading the manuscript from paper, it is very difficult to read the figures properly. The letter height of figures 5, 6, 7, 8, 10, 11 are too small (<1 mm, I had to use a magnifying glass). Besides the color of the figures could be also taking care of (yellow is very difficult to read).

Figure 5 is simplified (also following the suggestion of referee # 2) and the font of the remaining text is increased. The font size of axis titles/annotations and other labels in Figs. 6, 8, 10, 11 and 12 is increased, and the yellow colour in Figs. 6, 10 and 12 is replaced by a darker orange.

- For Fig 6, 10 and 12 the upper and lower panels have reversed wavenumber scales as compared to each other, which makes them difficult to compare.

This choice was made intentionally. In the top panel, the laser ramp is plotted in forward direction along the time axis; the wavenumber scale shown on the top axis is calculated, and the numbers are given as an indication and to illustrate that the wavenumber actually decreases along the laser scan. For the bottom panel, we chose to show the infrared absorption spectrum in the standard way against increasing wavenumber, i.e. with an increasing x axis scale. Also note that only a certain region (blue shading) from the top panel is shown here (absorption cannot be calculated for the ramp parts where the laser is off). We deemed this more logical and less confusing, and we do not see the added value of being able to visually draw direct lines from the ramp to the absorption spectrum. In addition, when the laser current ramp is linear as in Fig. 12, the relationship between ramp time and wavenumber is also not linear and one panel would have to be drawn with a distorted x-axis in order to provide for the direct comparison.

- For Fig. 6 lower panel, the difference between fit and data is very difficult to observe. I suggest also to show its difference in a separate panel (its residue)

In the revised manuscript, panels showing the residues are added to Figures 6 and 10.

**Additional references cited in this response:**

Gao, R. S., Ballard, J., Watts, L. A., Thornberry, T. D., Ciciora, S. J., McLaughlin, R. J., and Fahey, D. W.: A compact, fast UV photometer for measurement of ozone from research aircraft, Atmos. Meas. Tech., 5, 2201–2210, https://doi.org/10.5194/amt-5-2201-2012, 2012.

---

## Author Comment (AC2)

**Response to 'Comment on amt-2021-28', Anonymous Referee #2**

*We thank the anonymous referee for his or her constructive review of our AMICA manuscript. Some obvious mistakes, ambiguous phrasing and missing information was identified, which we try to remedy in the revised version of the manuscript. With respect to the suggestion to remove significant amounts of information on the more technical aspects of the instrument and on the two not fully operational channels, we make some changes but largely argue why we think this information is important and hopefully useful to at least some readers.*

*Original review comments are copied in black below, with our response in blue and changes/additions to the manuscript text in purple.*

The paper describes the AMICA instrument which uses OA-ICOS to measure a variety of trace species in the atmosphere using two distinct axes. Details are given on the mechanical, electrical, and optical configuration of the instrument as well as preliminary laboratory calibrations and descriptions of flight campaigns the instrument has participated in.

The paper is well written. However, the focus seems to be on the mechanical and electrical details (10 pages) rather than the optical setup, calibrations, and justifying the uncertainty numbers given in the paper (6 pages).

The strong focus on the mechanical and electrical details is owed to the fact that these really are the innovative aspects of the instrument that go beyond the state-of-the-art (the most important points being the bullets in Section 6).

Also following the Frans Harren's comments (cf. response to Frans Harren's review), we significantly expand the discussion on the calibration experiments and uncertainties in Section 4.1, and we better tie the numbers given in the abstract to this section.

Concerning calibrations, we plan further experiments to investigate long term drifts and to investigate the issues compromising the accuracy of our $CO_2$ measurement.

For example, there are several paragraphs that go into detail on the screws and washers used and how they were tightened, but little detail on the optical setup including the cavity mirrors. I am sure a lot of time was spent on the mechanical and electrical details and it's tempting to want to have the paper represent the time spent on the instrument development. While these details are important to provide the aircraft team when the instrument is integrated into those aircraft, it's not important or necessary to go into that much detail in an instrument paper.

It is true that the amount of detail devoted in the paper to different aspects of AMICA to some extent correlates with the amount of time spent on these during development. But as stated above, it also correlates with the amount of innovation that lies in the different details of this airborne instrument. In our opinion, it really is such innovative features that justify the publication of a scientific instrument paper, and we hope that they will prove useful to other instrument designers. Some of them, e.g. the "bolt & glue" enclosure and the two valve pressure regulation of the sampling system, are not necessarily limited or specific to the ICOS technique and thus may be of interest to wider audience.

Sizes and sources of screws, bolts and washers are given only where the choice of hardware type, material and size is critical in the context of stability and withstanding pressure gradients. Removing this information that is largely in parentheses would not significantly reduce the length of the paper and we prefer to leave this information in the paper.

With respect to some details of the electrical setup, we do agree that some information given is not necessary. Even for someone trying to build a 1:1 copy of AMICA, wire gauge and pin information are not necessary, as the former can be derived from the expected current flow and the latter is, to a large extent, arbitrary. Therefore, we remove these details from the paper. This mainly concerns the block diagram in Figure 5, the revised version of which is much simpler and more accessible.

The paper would read better if the authors spent as much time discussing the spectroscopic details of the instrument.

The OA-ICOS optical setup is adopted 1:1 from other mid-infrared analyzers available from ABB Los Gatos, in other words: with respect to the optics, we haven't done anything new in AMICA. By going into greater detail here, we could only repeat what was presented in existing publications, some of which go into great detail with respect to optical design and some of the caveats associated with it (respective references are given in the paper).

Nevertheless, the optical beam path is added to Figure 2 following Frans Harren's suggestion and more details on specific mirrors and lenses used in AMICA (see response to specific comment further below) are given in the revised manuscript.

Overall, I think this paper is a good fit for AMT and can be published after some major revisions are made to the sections discussing the spectroscopy and calibrations as well some other minor points that should be addressed. I would also suggest to the authors that much of that detail of the mechanical and electrical hardware be put into the supplementary section or removed.

More detailed information is added on the spectroscopy and optical setup and the section on calibrations and in particular the justification of the numbers is revised (see answers to specific comments below for the detailed changes applied).

As argued above, we refrain from removing or moving to the supplement details of the mechanical and electrical hardware because much of it is related to innovative aspects and features of the AMICA instrument.

One major change that I feel I need to highlight upfront is the abstract and text list three axes and associated molecules but only one of those axes has been characterized for all molecules. Even then, as I point out in the detailed comments, I have real concerns about the usability of some of the data, particularly the $CO_2$ and CO. Drifts in these molecules are larger or on the same magnitude as expected atmospheric changes and no in-flight calibration system is discussed.

At present, the AMICA $CO_2$ data are not used for scientific purposes and are not included in any campaign data files. The major issue currently compromising the retrieval of reliable $CO_2$ data from the AMICA spectra, namely the absorption at atmospheric mixing ratios being so strong that it significantly reduces the path length and potential uncertainties in the HITRAN line parameters, are already described in the manuscript (Sections 3.2 and 4.1) and supplement (Figure S5). As stated in Section 4.1, the use of calibration factors derived from regular calibrations can to some extent remedy these issues, but the uncertainty with such factors in place is much larger than that of other $CO_2$ instruments used during the campaigns. In the revised manuscript, we add the following text in Section 4.1 to clarify this:

"As a result of the spectral fitting issues and the additional uncertainty of the calibration factor, the error margins of the AMICA $CO_2$ data are currently on the order of a few ppm, much larger than those of other $CO_2$ instruments used during the field campaigns. Therefore, AMICA $CO_2$ measurements up to this point are not used for scientific purposes and are not part of any data files released

for the campaigns. We plan to verify and/or adjust the HITRAN parameters for the $CO_2$ line at 2050.60 $cm^{-1}$ in a future laboratory experiment with low $CO_2$ concentrations that do not reduce the effective path length at different pressures and temperatures, and then use them in the full fitting algorithm described by Sayres et al. (2009), where the effect on path length is mathematically represented."

We also expand the discussion on the Allen plots (Figure 9) and precision, and particularly discuss potential long term drifts:

"Because of the short time periods of 5 and 45 minutes used in these experiments, long term precision caused by instrumental drifts over longer time periods cannot be ruled out, and the observed behaviour at averaging times longer than about 200 s appears to point into that direction, although it is not conclusive. A measurement of the same gas standards with the current AMICA configuration over a period of several hours will be carried out in future to further investigate the susceptibility towards long term drifts. We also expect to achieve a further lowering of the σ curves in the future by further reducing electrical noise. For the OCS and CO observations made so far, respective precision estimates of 30 ppt and 3 ppb are made based on the higher two second value from the two experiments shown in Figure 9. For $CO_2$, both curves exceed 1 ppm for all averaging times, which is another reason (besides the issues described above) for currently not releasing AMICA $CO_2$ data for scientific use."

Besides these additions to the discussion in Section 4.1, the following text on issues related to ensuring data accuracy in absence of an in-flight calibration system is added at the very end of Section 3:

"In theory, these spectral fitting procedures avoid the need for frequent calibrations and in particular an in-flight calibration system that would substantially add to the instrument dimensions and weight, complicate air-worthiness and safety compliance certification, and lead to data gaps during calibration periods. There are, however, some caveats to this "calibration free" fitting:

- While absorption line parameters are constant by definition, they still need to be accurately known, and stated line uncertainties in HITRAN vary significantly.

- The absorption path length determination from the ring down fit needs to be precise and accurate.

- Precise line locking must ensure that the scanned wavelength scale is constant.

- The baseline must remain stable or at least well characterized by a mathematical function that can be fully included in the spectral fit.

- Cavity temperature and pressure need to be accurately known and constant over the time scale of the measurement.

While the last point is addressed by regular tests and, if needed, recalibration of the cavity pressure and temperature sensors (see Section 2.4), the former issues are specific to setup or absorption line and are discussed for each setup in Section 4. Clearly, for operational channels producing atmospheric data to be used for scientific purposes, validation of the complete system to detect potential systematic errors and to ensure data quality is done in the laboratory at regular intervals by measuring "zero air" as well as known standards. Some results of these experiments and a discussion of issues detected is given in Section 4."

For the second axis, $O_3/NH_3$, only ozone has been measured. $NH_3$ has never been observed even in lab and therefore I think it should be removed from the list of molecules.

It is true that we cannot currently measure $NH_3$ atmospheric concentrations. To make this even more clear, we add a note on "insufficient sensitivity" in the abstract, replace "can be measured" by "absorption lines exist" in the second sentence of Section 4.2, and mark $NH_3$ as not currently measured in the Table 3.

We prefer to leave the information that the $NH_3$ line falls within the spectral range of the 1030 cm⁻¹ in the paper because the measurement of $NH_3$ is a major target for future missions. For the SouthTRAC mission, we put our focus entirely on the realization of the $O_3$ measurement, but it should be noted that the 1030 cm⁻¹ channel is using the same laser/mirror/detector configuration as the Ammonia Analyzer available from Los Gatos (https://www.lgrinc.com/analyzers/overview.php?prodid=24&type=gas ) and the airborne Ammonia instrument described in the Leen et al. (2013) reference.

And in preliminary tests with the 1035 cm⁻¹ channel, we have recorded a $NH_3$ spectrum with this channel in the laboratory:

[Figure]

As in Figure 10 in the paper, the upper panel shows the detector signal for one laser ramp, and the lower panel shows the spectrum in absorption space. The blue line does not show a fit but rather a HITRAN based spectrum for the 57 ppb $NH_3$ mixing ratio in the standard that was used.

We chose not to include this spectrum in Figure 10 or otherwise show it in the paper because it was recorded while fine tuning the temperature and the ramp parameter setting of the laser prior to the SouthTRAC campaign, and the settings used are slightly different from what was used during campaign and the wavenumber scale for these settings is only approximate. Also, cavity pressure and temperature as well as some other housekeeping parameters had not been recorded to file at the time.

The third axis is a concept for which you have a laser, but no mirrors capable of measuring at atmospheric relevant concentrations. This should be removed from the manuscript.

Indeed, the third channel is more conceptual than operational. Nevertheless, there are two reasons why we would very much like to keep the short one-paragraph section 4.3 in the paper:

1. This configuration was actually flown during two campaigns. In retrospect, this proved too ambitious and something simpler (e.g. based on an existing Los Gatos Analyzer) may have been the better choice. But the channel was operational at least in the sense that infrared spectra over the nominal wavelength regions were recorded (and absorption at least for $H_2O$ at the ground was observed). We consider showing this information (including the measured spectra) as better than simply stating that we flew a second channel that measured spectra but wasn't sensitive enough.

> We also deem showing actual Cavity 2 spectra from two different configurations as important in the context of demonstrating the exchangeability of configurations.

2. We believe in the potential to make atmospheric measurements of $N_2O$, HCN and $C_2H_2$ in future with improved sensitivity. Mirrors of 0.9995 reflectivity at 3330 cm$^{-1}$ do exist, and more powerful ICLs in that wavelength range have become available (up to 25 mW, e.g. https://nanoplus.com/en/icl/). The conceptual description of this channel, the demonstration of spectral recording and the estimation of criteria required to realize atmospheric measurements in the current paper may be helpful in the process of moving this development forward.

Detailed Comments:

Line 14: I'll make more specific comments on this later, but it's not clear to me that the instrument can measure all these molecules at real atmospheric concentrations and with required precision or accuracy.

We add "with the aim" between "implemented" and to "measure" in this line to make it more clear that many of the the listed gases do not yet classify as AMICA data products for scientific use. Further below in the abstract, we explicitly give accuracy and precision only for the current "data products" OCS and CO:

> "For OCS and CO, data for scientific use have been produced with 5 % accuracy (15 % for CO below 60 ppb, due to additional uncertainties introduced by dilution of the standard) at typical atmospheric mixing ratios and laboratory measured 1-sigma precision of 30 ppt for OCS and 3 ppb for CO at 0.5 Hz time resolution."

This becomes obvious as you only list precision for one of the axes and say the others are in development (line 20). If they are not fully developed, they shouldn't be included in this paper.

Our reasons for including the other axes and information on the tests we have realized with them up to now are explained in response to the general comment above.

Line 19: Please give the time measurement for the precision as well as how many standard deviations. Is this 1-sigma, 1-second? 1-sigma, 2.5-seconds?

This information is added in the revised abstract:

> "… and laboratory measured 1-sigma precision of 30 ppt for OCS and 3 ppb for CO at 0.5 Hz time resolution."

Line 27: Suggest giving the typical flight speed of the aircraft and computing the resulting horizontal resolution. You do this later on in the paper but would be good in the abstract as well.

This information is added in the revised abstract:

> "Sample flow on the order of 1 SLM maintained by an exhaust-side pump limits the useful time resolution to about 2.5 s (corresponding to the average cavity flush time), equivalent to 500 m distance at a typical aircraft speed of 200 m s$^{-1}$."

Introduction: No motivation for measuring these particular molecules is given.

The motivation driving out channel selection is added in the introduction:

 "The initial choice of gases during the instrument development has been driven by the research group's scientific interest and objectives of initially planned missions. One trace gas of interest is carbonyl sulfide (OCS), the most stable and abundant reduced sulfur gas in the atmosphere and a precursor to stratospheric sulfate aerosol (Crutzen, 1976; Kremser et al., 2016) as well as a potential tracer for the important carbon cycle process of net primary production (Whelan et al., 2018). Using a prototype of both AMICA and the commercially available Los Gatos OCS Analyzer measuring near 2050 cm-1, OCS measurements have been conducted during field campaigns since 2014 mainly on research ships (Lennartz et al., 2017; Lennartz et al., 2020). In the wavelength region of the major OCS band in the infrared, carbon monoxide (CO), carbon dioxide ($CO_2$) and water vapour ($H_2O$) also absorb and are measured simultaneously by these analyzers. The attempt to measure hydrogen cyanide (HCN) and acetylene ($C_2H_2$) near 3332 cm-1 (where nitrous oxide, $N_2O$, also absorbs and can potentially be measured as an add-on) was motivated by their use as biomass burning tracers in the context of OCS (Notholt et al., 2003) and as pollution tracers in the Asian monsoon anticyclone (Park et al., 2008; Randel et al., 2010), the region of interest of two recent aircraft missions described further below. A cavity setup equivalent to the Los Gatos Ammonia ($NH_3$) Analyzer in the 1034 cm$^{-1}$ region was tested to measure ozone ($O_3$), which is abundant in stratospheric air expected to be sampled at high altitudes."

The new references are added to the reference list. To reflect this and to make it clear also in the introduction that not all gases were successfully *measured*, the sentence referring to Section 4 in the outline paragraph at the end of the introduction has also been reworded:

"Realized cavity setups at certain wavelength windows in the infrared aiming at the abovementioned target species are described in Section 4 that also includes results from laboratory tests and calibrations."

Lines 78 to 88: More detail is required to understand the optical configuration. Are lenses used to focus or collimate the beam emitted by the laser? Even if they are integrated into the laser housing, please give size, focal length or equivalent, part numbers. Details on the 90° deflection mirror: is it round, oval, flat? What cavity mirrors are used. Their R is given in Table 3, but you should also provide the ROC and manufacturer/coater. Collimating lenses on the detector side should also be listed with details.

More information on lenses and mirrors is added in Section 2.1:

"Each ICOS entity consists of a laser source (C7 in Figure 2 and Table 2), a 25.4 mm diameter round 90° deflection mirror with a protected silver coating, a 508 mm long cavity of 48 mm inner diameter with two 50.8 mm diameter concave high reflectivity mirrors with a 1 m radius of curvature (ABB Inc.), a 50.8 mm diameter 25 mm focal length aspheric/plano collimating lens (ZnSe for the 1035 and 2050 cm$^{-1}$ channels, Ge for the 3330 cm$^{-1}$ channel) and a detector (C8). The loosely collimated (by a refractive lens integrated in the laser mount) laser beam is aligned to enter the cavity slightly off-axis to minimize sensitivity to vibrations and to avoid interference patterns resulting from cavity resonance (Paul et al., 2001)."

Line: 82: You mention using piezo's but no information on why and how well they performed. Was fringing/etalons reduced by using the piezo's and if so by how much?

The following information on the purpose of the PZTs is added:

> "In addition, the position of each mirror is modulated by three piezoelectric transducers (PZTs, modulated by C12) to disrupt both intra and extra-cavity etalons that otherwise interfere with the spectroscopic analysis of small signals. PZTs have been found to reduce the magnitude of these etalons in Los Gatos analyzers to a varying degree, and the concept was adopted for AMICA without explicitly quantifying the magnitude of etalon reduction in this instrument."

Laser issues: Was laser feedback an issue? It often is in cavity systems, but I see no mention of it nor any typical optical solutions such as an isolator between the laser and cavity.

It is a significant advantage of OA-ICOS that the reflected beam is not returned directly into the laser which dramatically reduces the requirements for optical isolators between the laser and the cavity. This note is added in the revised manuscript:

> "Another advantage of the off-axis alignment is that the reflected beam is not returned directly into the laser which dramatically reduces the requirements for optical isolators between the laser and the cavity."

Was ASE an issue with these lasers? If the ASE is broad enough it can get around the cavity mirror coating and cause a few percent offset in the apparent light hitting the detector.

Amplified spontaneous emission from these lasers was not observed to produce a measurable offset in the spectra recorded by this analyzer.

Lines 159 to 212: I would move these sections to supplemental or remove. This is information that aircraft operations will want, but isn't needed in an instrument paper. For the main body of the paper, one paragraph could suffice for describing the how the instrument is mounted in the two aircraft.

We acknowledge that Section 2.3 contains rather detailed and specific information that may not be interesting or useful for all readers.

However, its *airborne* nature is one of the central features of the AMICA instrument, and as explained above, many of the innovative aspects of AMICA and therefore in this paper are related to solving some issues encountered when deploying an instrument in general and specifically an ICOS analyzer on aircraft.

We therefore find it important to go into some detail with respect to mounting the same instrument in a cabin rack as well as in an outside bay. And the reduction of vibrations is not only important in the context of aircraft safety, but also to protect the instrument interior and ensure that it works smoothly. Even though the off-axis alignment makes it very robust, more sizeable displacements (in the range of a few mm or a few degree angle) of the laser mounts or the deflection mirrors will significantly impair the measurement.

Line 232: Do you see any pressure oscillations due to the diaphragm pump?

Pressure oscillations on the time scales of recording (i.e. one or two seconds) are most likely caused by the response of the proportional valves (cf. response to the next comment below), but we cannot rule out pump effects with absolute certainty. We don't expect this to be significant, and we have observed no evidence for oscillations related to the nominal pump speed of 2400 rpm. Diaphragm pumps have successfully been used in most commercial Los Gatos analyzers and cavity pressure oscillations caused by these pumps are fast and small compared to the cavity volume and have never been found to be significant.

Line 244 and 249: Are the pressure regulation uncertainties the difference between the actual and the set-point or is pressure varying by these amounts over time as a result of either pressure fluctuations from the pump or fluctuations in the response of the proportional valves or noise on the pressure transducer?

The stated uncertainties represent 1-sigma precision. This is clearly stated in the revised version. We also add the following paragraph on the likely cause of pressure fluctuations:

> "The observed pressure fluctuations (given as 1-sigma standard deviations of pressure recorded at 0.5 Hz) most likely result from the response of the regulating valves. This is supported by the observations of higher fluctuations on the order of 2 – 3 hPa in preliminary tests using the larger valve at ~ 1000 hPa ambient pressure of ~1000 hPa, and reduced fluctuations at pressures below the lower set point when both valves remain fully open."

Pressure and temperature calibration: I see no mention of how pressure and temperature were calibrated? Were these devices purchased with calibration. What is the accuracy of that? The zeros of pressure transducers drift. Is that checked periodically?

The importance of accurate measurements of cavity pressure and temperature is stated in the new paragraph at the end of Section 3 (see above).

The following sentence is added in Section 2.4:

> "The pressure gauges (C4) are factury calibrated with an accuracy of 0.1 %. Recalibration before and after field campaigns is done in our lab against an absolute pressure baratron (MKS). Note that cavity temperature is measured with a thermistor that is calibrated in a glycol bath and accurate to about 50 mK."

Section 2.5: Again, an instrument paper does not need to include the jack numbers and which pins are connected, especially when it's all referenced to looking up what the letter/number designations are referring to. Most of this could be shortened to a few short paragraphs. Also avoid referring to everything by its letter/number designation. It is very confusing for someone not intimately used to the instrument.

Much of the "unnecessary" information was included in Figure 2 and is removed in the revised version.

With respect to the letter/number designations, we made the choice to use both the descriptive name – for clarity – and the designation – for the reader to be able to make the connection between the text and Figure 2/Table 2. We still believe that this is the most accessible way for most readers.

Lines 438 to 455: I'm confused to which method of fitting is used to fit the spectra shown in section 4.

Spectral fitting for the calibrations and precision tests described in Section 4, and for the SouthTRAC campaign data, was done using the fitting in absorption space. This is clarified by the flowing sentence added to Section 3:

> "The simplified fitting method in absorption space has been used for the spectra recorded during SouthTRAC (Section 5.1) and for the calibration experiments (Section 4)."

Line 462 to 464: These are individual ro-vibrational absorption lines, not bands.

This is, of course, correct, and the terminology is corrected throughout the revised manuscript.

Line 477: It's not clear what your resolution is to the problem you are describing with $CO_2$. Fitting an 80% deep line is challenging. Earlier in the paragraph you state that the fitting is done with full forward simulation which would therefore take into account the effects you describe later. But lines 475 to 477 seem to indicate that you are only fitting this with the approximation described by equations 2 and 3.

We have tried to use the full forward fitting for $CO_2$ for the M55 Geophysica spectra and some of the laboratory experiments. The resulting mixing ratios were closer to the true values (known standards or airborne measurements by another instrument) than with the absorption space fitting, but still not satisfactory. We suspect that one or more of the HITRAN parameters for the $CO_2$ line (that are based on theoretical calculations) are inaccurate to some degree. We plan to test this in experiments with low $CO_2$ (where $L_{eff} \sim L_0$) and hopefully adjust parameters when fitting increasingly higher known $CO_2$ concentrations. This is stated in the revised Section 4.1 (cf. response to the general comment above).

The fit itself presented in figure 6 is not very convincing. While the scale makes it hard to read there seem to be numerous differences between the data and your fit not just for the $CO_2$ line but for the other lines as well. It would be useful to plot the data zoomed in to the weaker lines and/or plot the difference between the data and the fit.

In the revised manuscript, panels showing the residues are added to Figures 6 and 10.

Line 478: You list it in the abstract but it would be good to give the precision numbers in this paragraph as well.

The same numbers given in the abstract are now given and further explained in Section 4.1.

Line 495: The Allen variance plot for OCS looks like what I would expect with noise averaging along the white noise line. For CO and $CO_2$ however, there is considerable structure in the data that does not average away which is why averaging past a few seconds does not improve the Allen variance plot. For $CO_2$ there seems to also be a longer drift at least equal to 2-sigma of the 1 second noise. No mention of this is made in the manuscript but these problems make the data unusable for scientific interpretation as the non-white variability (or drifting) is larger than expected atmospheric variability. A typical $CO_2$ instrument needs to measure sub-part per million to be useful; for CO it's part per billion. You state your group's interest is in OCS. What are the scientific questions that you intend to answer and what variability do you expect to see in OCS? Does the precision of this instrument allow you to answer those questions?

It is absolutely true that typical $CO_2$ instruments nowadays measure with sub-ppm precision. We clarify in the revised version that such precision is not currently achieved with AMICA data, which together with the described sensitivity and accuracy issues is the reason that no $CO_2$ data have been released for scientific use.

There also is no mention of in-flight calibration that could be used to correct the long term drifts seen in $CO_2$, though the short term drifts are also problematic. Typical flight instruments would have a gas standard deck that would periodically be used to check instrument operation and correct for slow drifts.

As stated above in response to the general comments, the discussion on $CO_2$ accuracy, precision and long term drifts in Section 4.1 is expanded in the revised version. Text on issues related to ensuring data accuracy in absence of an in-flight calibration system is added in Section 3.

Section 4.2: It seems like a lot of work needs to be done on this axis before it could be used for scientific investigation. You list $NH_3$ as a molecule but according to line 525 you have not observed it in flight nor

tested it in the laboratory. I don't think you should be listing it as a molecule you measure which was implied in the abstract.

As stated above in the response to the general comments, the 1034 cm$^{-1}$ channel is using the same laser/mirror/detector configuration as the Ammonia Analyzer available from Los Gatos (https://www.lgrinc.com/analyzers/overview.php?prodid=24&type=gas ) and the airborne Ammonia instrument described in the Leen et al. (2013) reference, and we have in fact been able to observe NH$_3$ absorption with this configuration in an experimental setup.

We therefore leave NH$_3$ in the list of target gases for this setup, but make it clear in the revised version that an NH$_3$ measurement with AMICA has not been operationally realized.

For ozone, you're spectral fit does not fit the data. All the lines in the spectra appear broader than your fit which would mean you are underestimating the concentration of ozone by the fit, yet no mention of this is given in the text when discussing why your fits underestimate the ozone mixing ratios. How much ozone is supposed to be in the spectra? It looks very noisy compared to the depths of the lines, but perhaps it's not that much ozone.

The following discussion on the obviously poor fit to the ozone spectrum is added to the first paragraph of Section 4.2:

> "The spectral fit does not closely reproduce the observe spectrum. First, significant absorption up to 0.01 between peaks points to either a bias in the used baseline or trace gas absorption by lines or bands that are not included in the HITRAN data base. Second, the observed O$_3$ absorption peaks are broader than the fitted peaks. Possible explanations include inaccurate HITRAN parameters for the O3 lines or cavity response broadening. Both, baseline offset and the broader peaks, will be further investigated in future laboratory experiments."

What is the line width of the laser being used?

The laser we use has a line width < 1 MHz, shown in several publications (e.g. Tombez et al., 2013).

Section 4.3: This section and references to this wavelength should be removed from the paper. You have not measured, as you state in section 4.3, any of these molecules at atmospheric concentrations, nor can you with the cavity optics you have. Even in the lab you have only put in one gas to show that a laser you purchased in lazing at the wavelength the company said it did. That is not new nor worthy of being published. Having it greatly detracts from the paper.

As stated above in the response to the general comments, we add this short section to demonstrate that two cavities in the same instrument were operational at least in the sense that they recorded infrared spectra, and because this conceptual description may actually help us to move the further development of this configuration forward.

It is made even clearer in the revised manuscript that this cannel is experimental rather than operational and does not measure the target molecules at atmospheric concentrations.

Line 558: It is hard for me to imagine that with reduced precision from that shown in section 4 that data from this campaign would be of scientific interest. Unless these data are being used in papers this whole section should be removed. It is clear that a lot of work has gone into the building and testing of this instrument. However, a published paper should not contain every thing that went wrong and didn't work and needed to be fixed.

At least the CO data from this campaign have been used in a publication (von Hobe et al., 2021). In that paper, precision was not so critical because the analysis involved a significant amount of averaging.

Also, the problems described in this Section represent a key step in optimizing or fine-tuning some of the innovative instrument features and in choosing components used in the current AMICA configuration. There were, in fact, quite a few more things that "went wrong" but are not included in the paper because these problems did touch any design issues.

Line 591: There is no von Hobe et al., 2020 in the references. Perhaps you meant 2021?

Indeed. This is corrected in the text.

Table 3: Bandwidth of the detector/preamp combination should be given. I'm assuming it's much faster than the ringdown time of the cavities?

The bandwidth, limited by the preamp, is 200 kHz. As this is the same for all channels used, it is added in the text in Section 2.2 rather than in Table 3.

Figure 5: I would move to supplemental as it's unnecessary detail for a paper.

We agree that Figure 5 contained a lot of unnecessary detail. While we prefer to leave it in the main paper, we have simplified the revised version, e.g. we removed the pinning and wire gauge information as well as some electrical component details.

Figures 6 and 10: These figures are hard to read. A third panel showing data minus fit would be useful. The vertical colored lines are not explained in the caption. It would be better for the x-axis of all the plots to be the same. The wavelength scale is going in the opposite direction in plot two compared to plot one making it hard to match up lines.

In the revised manuscript, panels showing the residuals are added to both Figures, and the vertical lines are explained in the caption.

The vertical lines were in fact included to help matching up the lines. Both (or now all three) panels using the same x-axis would be difficult and potentially confusing. We intentionally plotted the laser ramp in forward direction along the time axis in the top panel, showing the calculated wavenumber scale on the top axis only for guidance and to illustrate that wavenumber actually decreases along the laser scan. For the bottom panel, we chose to show the infrared absorption spectrum in the standard way against increasing wavenumber, i.e. with an increasing x axis scale. Also note that only a certain region (blue shading) from the top panel is shown here (absorption cannot be calculated for the ramp parts where the laser is off). We deemed this more logical and less confusing, and we do not see the added value of being able to visually draw direct lines from the ramp to the absorption spectrum (see also response to Frans Harren's review). In addition, when the laser current ramp is linear as in Fig. 12, the relationship between ramp time and wavenumber is also not linear and one panel would have to be drawn with a distorted x-axis in order to provide for the direct comparison.

All figures: Axis labels and numbers are hard to read. Please enlarge the font.

The font size of axis titles/annotations and other labels in Figs. 6, 8, 10, 11 and 12 is increased.

Figure 13: The pressure measurements show fast oscillations that I would think complicate and add uncertainty to the measurements. Is this from ringing in your pressure control algorithm or feedback between

the two solenoid valves? As the cavity pressure and temperatures are the most important for fitting the data, I would suggest just showing those and perhaps the enclosure temperature. Or if you want to show all of these making more sub plots so that the details of the measurement can be seem. For example, do you see a similar ringing of temperature in the cavity?

A revised Figure 13 is divided into more subplots. Cavity temperatures in particular are better resolved, and no 'ringing' is observed.

The pressure ringing is indeed caused by the solenoid valve control response, as is now explained in Section 2.4 (cf. above).

**Additional references cited in this response:**

Tombez, L., Schilt, S., Di Domenico, G., Blaser, S., Muller, A., Gresch, T., Hinkov, B., Beck, M., Faist, J., and Hofstetter, D.: Physical Origin of Frequency Noise and Linewidth in Mid-IR DFB Quantum Cascade Lasers, in: OSA Technical Digest (online), CLEO: 2013, San Jose, California, 2013.